# Reorganization of the flagellum scaffolding induces a sperm standstill during fertilization

**Martina Jabloñski[1], Guillermina M Luque[1], Matias Gomez Elias[1], Claudia Sanchez Cardenas[2], Xinran Xu[3], Jose L de La Vega Beltran[2], Gabriel Corkidi[4], Alejandro Linares[2,5], Victor Abonza[2,5], Aquetzalli Arenas-Hernandez[6], María DP Ramos-Godinez[7], Alejandro López-Saavedra[7,8], Dario Krapf[9], Diego Krapf[3], Alberto Darszon[2], Adán Guerrero[2,5]\*[†], Mariano G Buffone[1]\*[†]**

[1]Instituto de Biología y Medicina Experimental, Consejo Nacional de Investigaciones Científicas y Técnicas (IBYME-CONICET), Buenos Aires, Argentina; [2]Departamento de Genética del Desarrollo y Fisiología Molecular, Instituto de Biotecnología, Universidad Nacional Autónoma de México, Morelos, Mexico; [3]Department of Electrical and Computer Engineering and School of Biomedical Engineering, Colorado State University, Fort Collins, United States; [4]Laboratorio de Imágenes y Visión por Computadora, Instituto de Biotecnología, Universidad Nacional Autónoma de México, Morelos, Mexico; [5]Laboratorio Nacional de Microscopía Avanzada, Instituto de Biotecnología, Universidad Nacional Autónoma de México, Morelos, Mexico; [6]Departamento de Microscopía Electrónica, Instituto Nacional de Cancerología, Mexico, Mexico; [7]Unidad de Aplicaciones Avanzadas en Microscopía, Instituto Nacional de Cancerología Unidad de Investigación Biomédica en Cáncer, Mexico, Mexico; [8]Tecnologico de Monterrey, Escuela de Medicina y Ciencias de la Salud, Mexico City, Mexico; [9]Instituto de Biología Molecular y Celular de Rosario (IBR), Consejo Nacional de Investigaciones Científicas y Técnicas (CONICET), Universidad Nacional de Rosario (UNR), Rosario, Argentina

**\*For correspondence:**
adan.guerrero@ibt.unam.mx (AG);
mgbuffone@ibyme.conicet.gov.ar (MGB)

[†]These authors contributed equally to this work

## eLife assessment

This **important** work substantially advances our understanding of sperm motility regulation during fertilization process by uncovering the midpiece/mitochondria contraction associated with motility cessation and structural changes in the midpiece actin network as its mode of action involved. The evidence supporting the conclusion is **solid**, with rigorous live-cell imaging using state-of-the-art microscopy, although more functional analysis of the midpiece/mitochondria contraction would have further strengthened the study. The work will be of broad interest to cell biologists working on the cytoskeleton, mitochondria, cell fusion, and fertilization.

**Abstract** Mammalian sperm delve into the female reproductive tract to fertilize the female gamete. The available information about how sperm regulate their motility during the final journey to the fertilization site is extremely limited. In this work, we investigated the structural and functional changes in the sperm flagellum after acrosomal exocytosis (AE) and during the interaction with the eggs. The evidence demonstrates that the double helix actin network surrounding the mitochondrial sheath of the midpiece undergoes structural changes prior to the motility cessation. This structural

modification is accompanied by a decrease in diameter of the midpiece and is driven by intracellular calcium changes that occur concomitant with a reorganization of the actin helicoidal cortex. Midpiece contraction occurs in a subset of cells that undergo AE, and live-cell imaging during in vitro fertilization showed that the midpiece contraction is required for motility cessation after fusion is initiated. These findings provide the first evidence of the F-actin network's role in regulating sperm motility, adapting its function to meet specific cellular requirements during fertilization, and highlighting the broader significance of understanding sperm motility.

## Introduction

Sperm motility is required for arrival at the site of fertilization and to penetrate the different layers surrounding the egg. The temporal regulation of sperm motility involves the concerted action of multiple cell structures to enable fertilization. The initial motility of ejaculated sperm is characterized by a linear progressive movement as the cells traverse a long distance within the female reproductive tract. However, at one point, before reaching the oocyte, this initial progressive motion must be changed to a vigorous non-progressive mode of motility called hyperactivation (*Demott and Suarez, 1992*). During that migration, most of mouse sperm undergo acrosomal exocytosis (AE), which takes place in the upper segments of the oviduct, before the sperm directly interact with the egg or its surrounding layers (*Hino et al., 2016*; *La Spina et al., 2016*; *Muro et al., 2016*). This exocytic event is critical because proteins involved in this process are rearranged in preparation for fusion (*Inoue et al., 2005*; *Noda et al., 2020*). A second dramatic change in motility is subsequently required for efficient sperm-egg fusion. During this event, which has not been studied in detail so far, sperm completely stop moving (*Gaddum-Rosse et al., 1984*; *Gaddum-Rosse et al., 1982*; *Ravaux et al., 2016*). The cease in sperm motility is considered as an indicative marker of an effective fusion between gametes, given that it is necessary to complete the attachment and fusion between sperm and eggs. Fusion itself is a complex event mediated by several proteins identified using loss-of-function strategies (*Deneke and Pauli, 2021*; *Siu et al., 2021*). Nevertheless, the available information about how sperm regulate their motility during the final journey to the fertilization site and during the interaction with the female gamete is extremely limited. In this final journey, most sperm cells migrate to the ovulated eggs after AE, and very little is known about the motility state of those sperm and about the molecular mechanism in charge of the motility arrest.

The regulation of sperm motility is achieved by balancing force transduction and the mechanical properties of the flagellum. Both effects are governed by the flagellum cytoskeleton, which consists of two major components: a microtubule-based axoneme, located at the flagellum axis, and actin filaments around it (*Otani et al., 1988*). The midpiece and the principal piece also contain the outer dense fibers (ODFs) and the fibrous sheath that have traditionally been referred to as part of the sperm cytoskeleton. Recently, it was found that the three-dimensional (3D) organization of polymerized actin in the flagellum midpiece of murine sperm forms a unique double helix arrangement accompanying mitochondria (*Gervasi et al., 2018*). This spatial distribution does not extend into the principal piece, where actin is uniformly distributed between the axoneme and the plasma membrane. Across all investigated cells, this double helix consisted of exactly 87 gyres with a pitch of $244 \pm 1$ nm, yielding a total midpiece length in mice of $21.2 \pm 0.3$ μm (*Gervasi et al., 2018*). Such accurate control of sizes is not typical in live cells given that reactions are prone to stochastic effects and entropy leads to large cell-to-cell fluctuations. Thus, the precise control of the actin helix comes at a substantial regulatory cost for sperm cells. Nevertheless, the function of this specialized structure of actin filaments is largely unknown.

In this work, we find definite evidence for the role of the helical actin structure in the final stage of motility regulation, when sperm dramatically shift from being hyperactivated to a practically immotile stage. We have investigated the structural changes in the actin cytoskeleton of the sperm flagellum after AE and during the interaction with the eggs. By using a combination of single-cell imaging and super-resolution microscopy methods, our results demonstrate that the helical structure of polymerized actin undergoes a radical structural change at the time of sperm-egg fusion. Further, we uncover that this process is triggered by a substantial increase in intracellular calcium ($[Ca^{2+}]_i$). This actin-dependent signaling pathway promotes a decrease of the sperm midpiece diameter and motility arrest, both of which are found to be required to complete the fusion process during fertilization.

## Results

### AE promotes a cessation of motility in a subset of sperm cells

After AE, sperm need to accomplish migration to the ampulla, reach unfertilized eggs, penetrate the cumulus matrix and the zona pellucida, and finally, undergo fusion with the oocyte (*Yanagimachi, 1994*). All these processes share the need to modulate sperm motility. Little is known about the regulation of motility after the occurrence of AE or prior to the fusion with the oocyte.

To simultaneously monitor the acrosomal status and sperm motility in live cells, transgenic mice whose sperm express enhanced green fluorescent protein (EGFP) in the acrosome and red fluorescent protein (DsRed2) in the mitochondria were used (*Figure 1A*, *Figure 1—video 1*). EGFP-DsRed2 sperm were immobilized at the head on laminin-coated coverslips, while still allowing free flagellar movement (*Figure 1B*, *Figure 1—video 2*). While the presence of the acrosome is monitored using the EGFP fluorescence signal, motility was assessed through the movement of the midpiece (DsRed2 fluorescence), which was beating in and out of the imaging plane (*Figure 1B*).

The majority of acrosome-intact sperm were able to move. Interestingly, *Figure 1B and C* show the coexistence of two populations of sperm that underwent AE (cells lacking EGFP signal) induced upon addition of progesterone, a physiological trigger of AE. Some of the acrosome reacted sperm moved normally (34.1 ± 3.7%, n = 235), whereas the majority of them remained immotile (65.9 ± 6.2%, n = 2350) (*Figure 1B and C*, upper and lower panels, respectively, *Figure 1—video 2*).

The flagellar beat cycle encompasses a self-regulatory mechanism that receives feedback from molecular and mechanical signals (*Inaba and Shiba, 2018*). The complete abortion of flagellar beat cycle observed in *Figure 1B and C* might be indicative of a stimulus provided by or occurring concomitant with AE. To comprehend the connection between AE and motility, we hypothesized the existence of a mechanical change in the flagella as a result of the AE. For this reason, AE was studied in the presence of FM4-64 fluorescent dye (*Sánchez-Cárdenas et al., 2014*) to visualize structural changes at the plasma membrane, which can occur at macroscopic scales observed as an alteration in the shape of the flagellum, or at mesoscopic scales, monitored by local changes in FM4-64 fluorescence occurring at the vicinity or within the plasma membrane. EGFP-DsRed2 sperm were stimulated with progesterone and their 'motility' behavior was recorded for 5 min in the presence of FM4-64. The loss of EGFP fluorescence in the acrosome of transgenic mice correlated with a noticeable increase of FM4-64 fluorescence in the head, as shown in *Figure 1D* (see panels 0′ and 3′, *Figure 1—video 3*). In addition, we noticed that sperm that underwent AE later showed an increase in the FM4-64 fluorescence intensity in the flagellum as well (see panel 14′of *Figure 1D*, *Figure 1—video 3*).

A more detailed analysis of individual cells demonstrated changes in FM4-64 fluorescence according to their motility (*Figure 1—video 4*). *Figure 1E* shows a motile acrosome-intact sperm (a control case), where the fluorescence levels of EGFP and FM4-64 remained constant during the entire experiment (Pattern I). Interestingly, sperm that lost their acrosome (no EGFP and high FM4-64 fluorescence in the sperm head) and continued moving, presented low levels of FM4-64 fluorescence in the midpiece (*Figure 1F*, Pattern II). On the other hand, acrosome-reacted sperm that remained immotile showed a significant rise in the FM4-64 midpiece fluorescence (*Figure 1G*, Pattern III). Noteworthy, both subsets of cells (with and without motility) displayed the expected increase in FM4-64 fluorescence in the sperm head.

To deepen our understanding on how sperm lose motility during AE, we designed a sperm-tracking system that can also monitor changes in fluorescence in moving cells (*Figure 1—figure supplement 1B and C*). Three parameters were then assessed in real time, namely, the beat frequency of the flagellum (by tracking it on bright field images, or the DsRed2 channel, *Figure 1—figure supplement 1A*), the status of the acrosome (EGFP signal), and changes occurring at or within the membrane in the midpiece (FM4-64 signal). *Figure 1—figure supplement 1B* shows that sperm that did not undergo AE remained motile with a stable beat frequency and low FM4-64 fluorescence over the recording time. In stark contrast, after AE, cells gradually diminished the beating frequency until a complete arrest is observed, while a gradual increase in FM4-64 fluorescence in the midpiece is observed (*Figure 1—figure supplement 1C*).

To evaluate the relationship between FM4-64 and viability, a vital dye Sytox Blue was used in imaging flow cytometry experiments. Fluorescence intensities in non-capacitated sperm stimulated with ionomycin (*Figure 1—figure supplement 1D*) were determined and two populations with Sytox Blue signals were clearly distinguished (Sytox+ and Sytox-), enabling the discernment between live

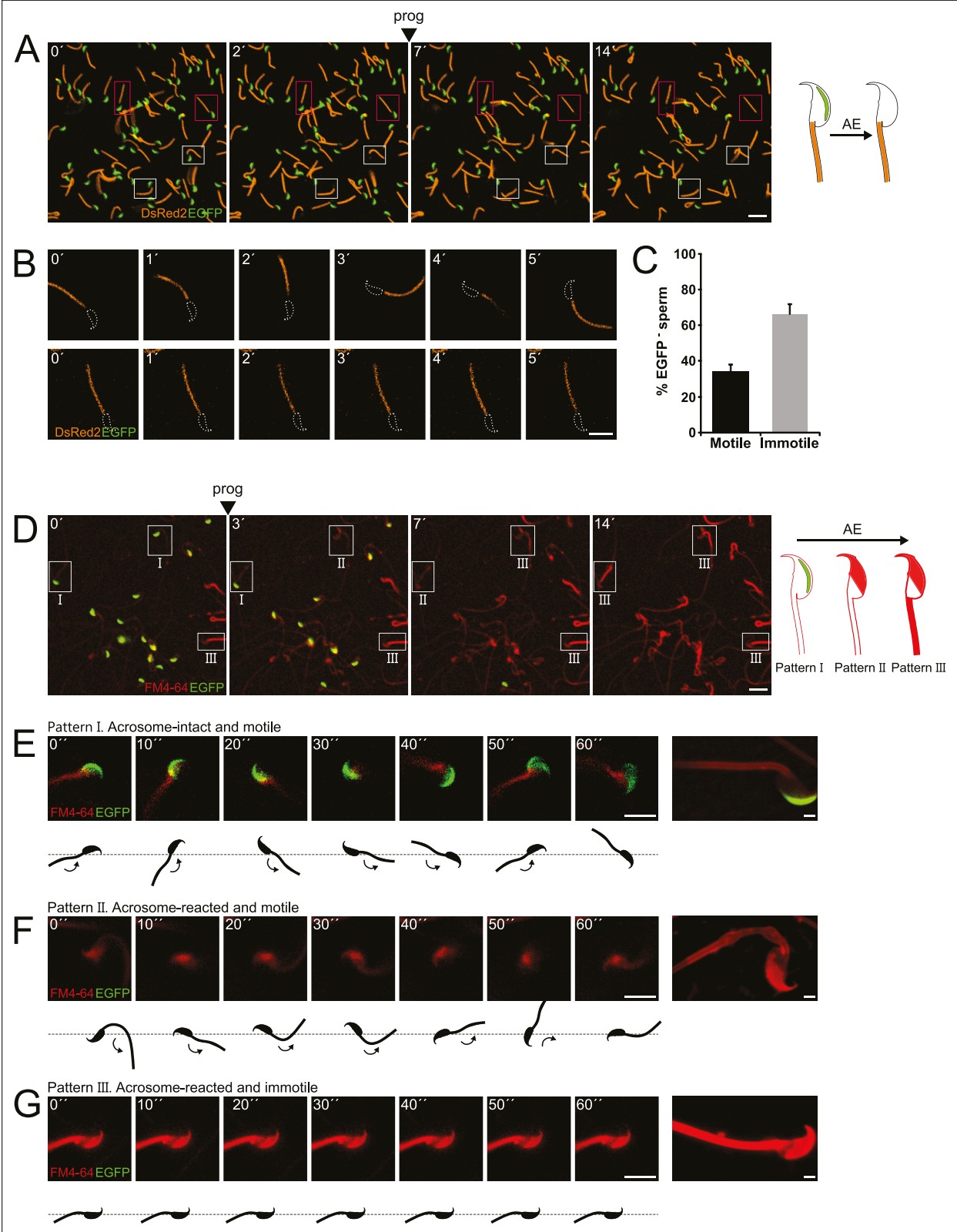

**Figure 1.** Sperm motility loss and FM4-64 fluorescence dynamics in acrosome-reacted transgenic EGFP-DsRed2 sperm. (**A**) Representative time series of transgenic EGFP-DsRed2 sperm attached to concanavalin A-coated coverslips, with acrosomal exocytosis (AE) induced by 100 μM progesterone. White squares indicate cells with spontaneous AE (prior to induction), while pink squares highlight cells with progesterone-induced AE. A schematic representation of AE in this transgenic model is shown on the right side of the panel. Scale bar = 20 μm. (**B**) Representative time series of transgenic

*Figure 1 continued on next page*

*Figure 1 continued*

EGFP-DsRed2 sperm that have already experienced AE, attached to laminin-coated coverslips. The upper panel displays a cell with motility after AE, and the lower panel shows an immotile cell. DsRed2 is presented in orange, and EGFP in green. Scale bar = 10 µm. (**C**) Quantification of motile and immotile acrosome-reacted sperm (EGFP-). A total of 235 cells were counted across at least three independent experiments. (**D**) Representative time series of transgenic EGFP-DsRed2 sperm stained with 10 µM FM4-64 and attached to concanavalin A-coated coverslips, with AE induced by 100 µM progesterone. White squares indicate cells exhibiting patterns I, II, or III after progesterone induction. Scale bar = 20 µm. A schematic representation of AE in this transgenic model stained with FM4-64 is shown on the right side of the panel. (**E–G**) Representative images of capacitated transgenic EGFP-DsRed2 sperm stained with 10 µM FM4-64. Panel (**E**) displays an acrosome-intact, motile sperm (Pattern I), (**F**) shows an acrosome-reacted sperm with motility and low FM4-64 midpiece fluorescence (Pattern II), and (**G**) presents an acrosome-reacted sperm with no motility and high FM4-64 midpiece fluorescence (Pattern III). In all three cases, cells were induced with 100 µM progesterone. Scale bar = 10 µm. Enlarged images of each pattern are shown on the right panel. Scale bar = 2 µm. Representative images from at least five independent experiments are displayed.

The online version of this article includes the following video and figure supplement(s) for figure 1:

**Figure supplement 1.** Gradual decrease in flagellar beat frequency following acrosomal exocytosis.

**Figure 1—video 1.** Representative movie of transgenic EGFP-DsRed2 sperm attached to concanavalin A-coated coverslips, with acrosomal exocytosis (AE) induced by 100 µM progesterone as indicated with Prog.
https://elifesciences.org/articles/93792/figures#fig1video1

**Figure 1—video 2.** Representative movie of transgenic EGFP-DsRed2 sperm that have already experienced acrosomal exocytosis (AE), attached to laminin-coated coverslips.
https://elifesciences.org/articles/93792/figures#fig1video2

**Figure 1—video 3.** Representative movie of transgenic EGFP-DsRed2 sperm stained with 10 µM FM4-64 and attached to concanavalin A-coated coverslips, with acrosomal exocytosis (AE) induced by 100 µM progesterone as indicated with Prog.
https://elifesciences.org/articles/93792/figures#fig1video3

**Figure 1—video 4.** Representative movies of capacitated transgenic EGFP-DsRed2 sperm stained with 10 µM FM4-64.
https://elifesciences.org/articles/93792/figures#fig1video4

and dead sperm. Interestingly, the upper right panels (Sytox Blue+/FM4-64 high) consistently show a positive correlation between FM4-64 and Sytox Blue. Nonetheless, the lower panels (Sytox Blue-) show no correlation with FM4-64 fluorescence, indicating that this population can exhibit either low or high FM4-64 fluorescence. Single-cell examples are shown, where the four categories are represented: dead sperm with low FM4-64 fluorescence (Sytox Blue+/FM4-64 low), dead sperm with high FM4-64 fluorescence (Sytox Blue+/FM4-64 high), live sperm with low FM4-64 fluorescence (Sytox Blue-/FM4-64 low), and live sperm with high FM4-64 fluorescence (Sytox Blue-/FM4-64 high). Therefore, while the FM4-64 signal alone is not a definitive marker for either AE or cell death, it is crucial to use additional viability assessments, such as Sytox Blue, to accurately differentiate between live and dead sperm in studies of AE and sperm motility. Cell viability was always considered, as any imaged sperm was chosen based on motility, indicated by a beating flagellum. The determination of whether selected sperm die during or after AE remains to be elucidated. The results presented in *Figure 1* and *Figure 1—figure supplement 1B and C* show examples of motile sperm that experience an increase in FM4-64 fluorescence.

Altogether, these experiments demonstrate that AE promotes a cease of motility in a subset of sperm cells, which coincides with an increase in FM4-64 fluorescence in the midpiece.

## AE is followed by a decrease in the midpiece diameter

Live-cell super-resolution microscopy was used to understand the structural changes occurring in the midpiece. *Figure 2A* shows a wide-field image of a sperm midpiece stained with FM4-64 (left) and analyzed using super-resolution radial fluctuations (SRRF, right) (*Gustafsson et al., 2016*). AE dynamics was monitored by FM4-64 fluorescence in the head (*Figure 2B*, upper-right insets). A decrease in the midpiece diameter over time was observed following AE either being spontaneous, induced by a physiological agonist (progesterone), or induced by a non-physiological surge of Ca$^{2+}$ (ionomycin) (*Figure 2B and C* and *Figure 2—video 1*). In the negative control (no AE), the diameter remained unchanged. This phenomenon was also observed using mean shift super-resolution (MSSR) (*Torres-García et al., 2022*) together with other plasma membrane probes of different molecular structure, such as Memglow 700 (*Figure 2—figure supplement 1A*), Bodipy-GM1 (*Figure 2—figure supplement 1B*), and FM1-43 (*Figure 2—figure supplement 1C*). In all three cases, a decrease in the midpiece diameter was observed after AE.

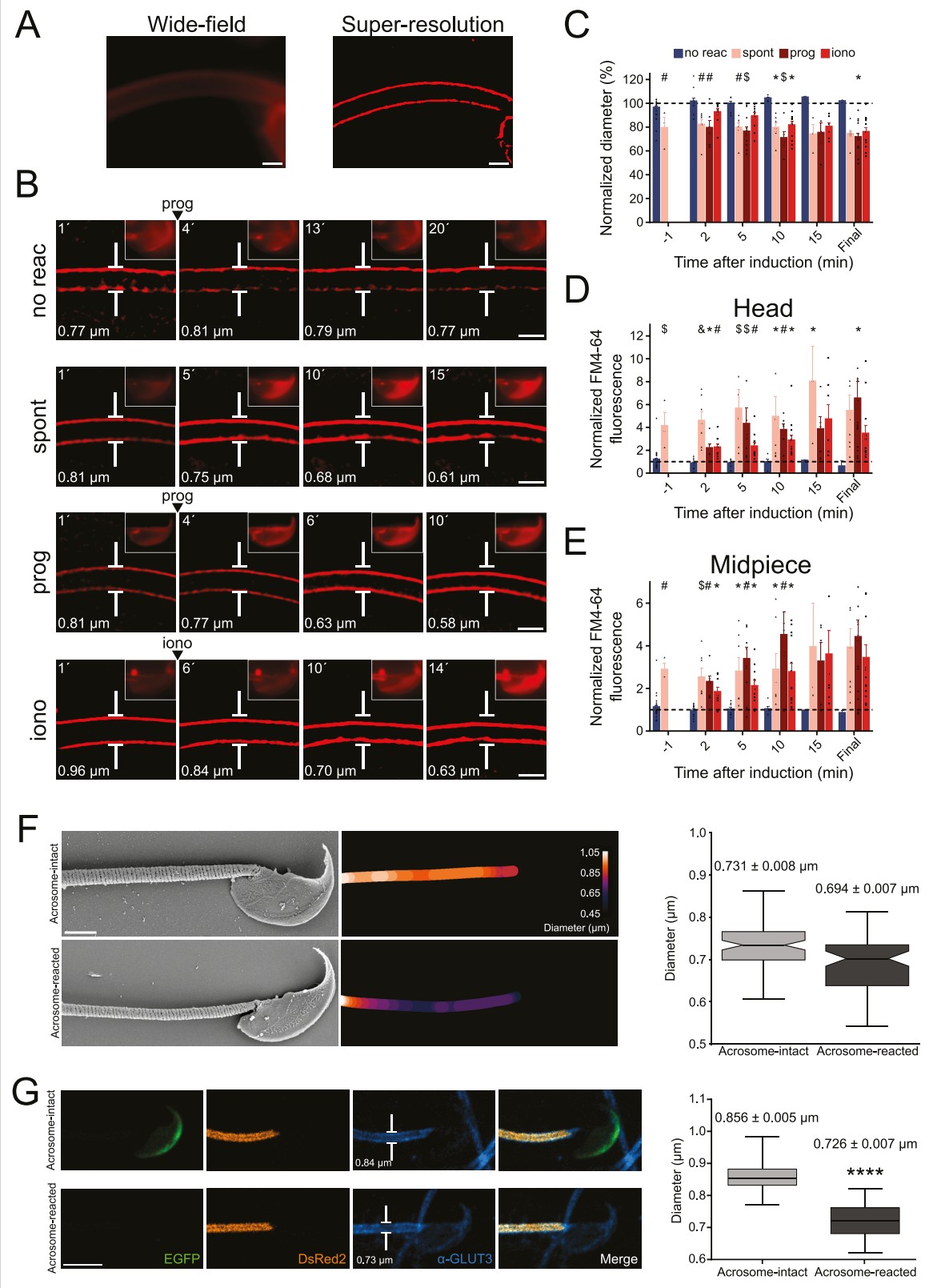

**Figure 2.** Midpiece contraction coincides with the onset of acrosomal exocytosis (AE). (**A**) Left panel displays a wide-field fluorescence image of capacitated CD1 sperm membrane stained with 0.5 μM FM4-64, while the right panel shows its super-resolution radial fluctuations (SRRF) reconstruction. Scale bar = 1 μm. (**B**) Representative time series of sperm midpiece with no AE (no reac), spontaneous exocytosis (spont), progesterone (prog, 100 μM), and ionomycin-induced (iono, 10 μM) exocytosis, respectively. Following acquisition, images were analyzed using SRRF. Insets in the sperm head show

*Figure 2 continued on next page*

*Figure 2 continued*

wide-field images of AE. The midpiece diameter value is displayed in the bottom-left corner for each time point. Scale bar = 1 μm. (**C**) Quantification of midpiece diameter changes for each experimental group across time. Data are presented as a percentage of the initial diameter value before induction for each cell. (**D, E**) Quantification of FM4-64 fluorescence in the sperm head and midpiece, respectively, for each experimental group across time. Data are presented as times of increases compared to initial fluorescence before AE induction. *$p<0.05$; #$p<0.01$; $$p<0.001$, and &$p<0.0001$ compared to the non-reacted group. A nonparametric Kruskal–Wallis test was performed in combination with Dunn's multiple-comparisons test. Representative images of at least five independent experiments are shown, with 36 cells analyzed. (**F**) Comparison of the midpiece architecture in acrosome-intact (AI, upper panel) and acrosome-reacted (AR, lower panel) sperm using scanning electron microscopy. Representatives images are shown, middle panels show quantification of these images whereas the left panel shows the quantification of all replicates. Data is presented as mean ± SEM, Kruskal–Wallis test was employed, $p=0.013$ (AI n = 85, AR n = 72). Scale bar = 2 μm. (**G**) Capacitated transgenic EGFP-DsRed2 sperm were induced by 100 μM progesterone. Cells were fixed and immunostained against α-GLUT3 in order to see the plasma membrane in the midpiece. Representative images of at least two independent experiments are shown. Left panel shows quantification of midpiece diameter in acrosome-intact and acrosome-reacted EGFP-DsRed2 sperm. Data is presented as mean ± SEM. A nonparametric Mann–Whitney test was performed, ****$p<0.001$ (AI n = 84, AR n = 47). Scale bar = 5 μm.

The online version of this article includes the following video and figure supplement(s) for figure 2:

**Figure supplement 1.** Correlation between FM4-64 increase and midpiece contraction.

**Figure 2—video 1.** Representative movie of capacitated CD1 sperm stained with 0.5 μM FM4-64 64 and attached to concanavalin A-coated coverslips with spontaneous acrosomal exocytosis (AE).

https://elifesciences.org/articles/93792/figures#fig2video1

The FM4-64 fluorescence intensity in the head and in the midpiece (*Figure 2D and E*) was also assessed. As expected, the increase in FM4-64 fluorescence in the head occurs after AE. Furthermore, in agreement to what was observed in *Figure 1*, fluorescence intensity in the midpiece also increased over time in sperm that underwent AE (*Figure 2D and E*).

To confirm these results, two alternative methods to visualize the change in sperm midpiece diameter were used. In neither of them, a membrane dye was used. First, indirect immunofluorescence to detect a membrane protein (GLUT3) was performed. As shown in *Figure 2G*, a decrease in the midpiece diameter was also observed. Second, scanning electron microscopy as used to evaluate the midpiece in acrosome-intact or reacted sperm (*Figure 2F*). The overall diameter of the midpiece in acrosome-intact sperm was larger than in acrosome-reacted sperm, with measurements of 0.731 ± 0.008 μm and 0.694 ± 0.007 μm, respectively. Overall, we have confirmed using three different approaches that AE is followed by a midpiece diameter.

## The contraction of the midpiece initiates in the proximal part of the flagellum

To investigate whether the contraction of the midpiece is triggered at a random location or in a particular region of the flagellum, kymographs were used. A kymograph allows visualization of dynamical aspects of a given phenomenon in a single figure, where the temporal dimension is expressed as an axis of the image through a spatial-temporal transformation of the dataset. Two types of kymographs were used (*Figure 3A*). First, super-resolution microscopy kymographs were built to monitor dynamic changes in the diameter of the flagellum along the midpiece. Second, to investigate the number of contraction sites, fluorescence kymographs were made from diffraction-limited images to observe changes in FM4-64 fluorescence over time and across the entire midpiece. Interestingly, *Figure 2— figure supplement 1D* shows a significant negative correlation between the midpiece diameter and the FM4-64 fluorescence, unveiling a tool to obtain an approximate value of the midpiece diameter without the need of super-resolution microscopy.

Three experimental subsets that underwent AE were assessed: spontaneous, induced by progesterone, and induced by ionomycin. To scrutinize randomness of focus-driven contraction, the midpiece was segmented in three regions (*Figure 3A*): proximal (near the neck), central, and distal (near the annulus). In these groups, it was then evaluated whether cells presented one or more foci of contraction.

*Figure 3B* shows a cumulative study, which indicates that contraction is preferentially initiated at the proximal part of the midpiece, regardless of being progesterone-induced or spontaneous AE. When ionomycin was used as an agonist of AE, the contraction began randomly in any segment of the midpiece. *Figure 3C and D* show two examples of flagellar dynamics for progesterone-induced

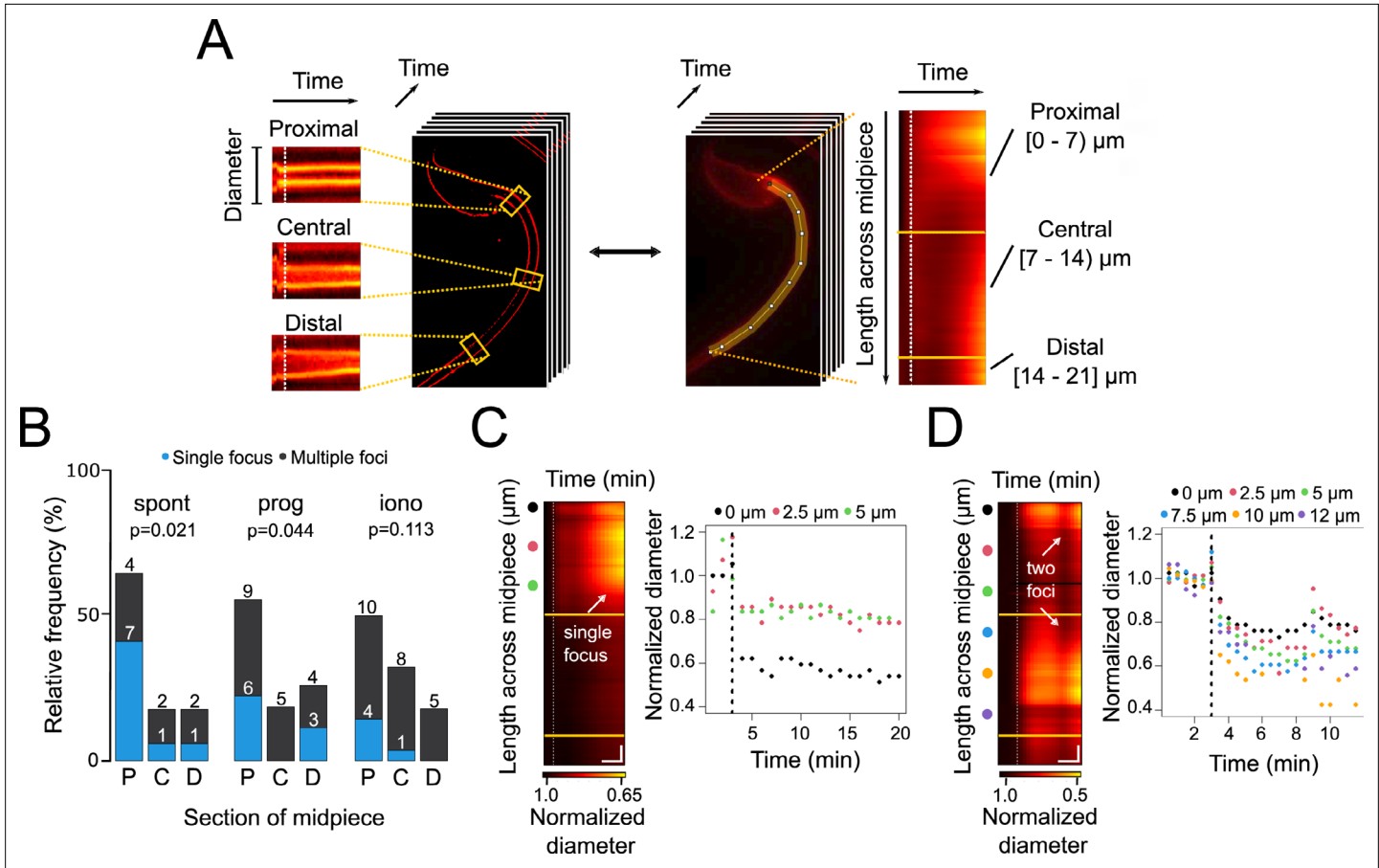

**Figure 3.** Contraction initiation preferentially occurs near the head-midpiece junction. (**A**) Schematic diagram illustrating the generation of super-resolution kymographs from super-resolution radial fluctuations (SRRF)-processed images. Crosslines are drawn every 2.5 µm through the sperm midpiece, and the ImageJ Kymograph builder plug-in is used to create kymographs. The x-axis represents time, and the y-axis shows diameter changes. For wide-field images, a line along the midpiece is drawn to create fluorescence kymographs, with the y-axis representing midpiece length. Three sections of the midpiece are defined: proximal (0–7 µm), central (7–14 µm), and distal [14–21 µm]. (**B**) Relative frequency graph displaying the distribution of the initiation sites for midpiece contractions in sperm with spontaneous exocytosis (spont), progesterone-induced (prog, 100 µM) exocytosis and ionomycin-induced (iono, 10 µM) exocytosis, respectively. The x-axis indicates the midpiece section where the contraction begins: proximal (P), central (C), or distal (D). A chi2 test was performed using the R language environment. (**C, D**) Representative contraction kymographs and diameter measurements for progesterone-induced (100 µM) AE with one or two contraction initiation sites, respectively. In contraction kymographs, yellow lines demarcate midpiece sections, and colored spots indicate where super-resolution kymographs were created. Both kymograph and diameter measurement graphs display a dotted vertical line marking the induction point. For (**C**), horizontal scale bar = 5 min and vertical scale bar = 1 µm and for (**D**), horizontal scale bar = 3 min and vertical scale bar = 1 µm. Data from at least five independent experiments are shown, with 36 cells analyzed.

sperm, which either presented one or two foci of contraction. In both cases, the diameter decreased along the whole midpiece, seen as a transition from dark to bright red colors in *Figure 3C and D*. Noteworthy, focal points of contraction were consistently observed: that is, a case with a single focus (*Figure 3C*), or a case with two foci of contraction (*Figure 3D*). *Figure 2—figure supplement 1E and F* show ionomycin-induced sperm that present one or more foci. It is observed in the normalized diameter kymograph that the flagellum presents a single focus between the measurements of 7.5 µm and 10 µm (*Figure 2—figure supplement 1E*, left panel). This is reflected in the right panel in the light blue and orange dots, which are the actual diameter measurements that decreased first and most (*Figure 2—figure supplement 1E*). A representative behavior within this experimental group is represented in *Figure 2—figure supplement 1F*, where the entire midpiece contracted (all its sections simultaneously). This is observed in the color scale of the normalized diameter kymograph and coincides with the real measurements of the diameter in the right panel, which decreased their value. Lastly, *Figure 2—figure supplement 1G and H* show examples of spontaneous AE with one or two foci of contraction, respectively. In particular, in *Figure 2—figure supplement 1H*, a first focus

of contraction between 0 and 5 µm that coincided with the diameter measurements of the right panel that decreased their value first and to a greater extent (black, pink, and green dots). Although the entire midpiece contracted, a second focus of contraction appeared around the measurements of 10–15 µm and coincidentally, these are the points with the strongest decrease in the diameter (orange, purple, and red points).

## The reduction of the midpiece diameter occurs concomitantly with an $[Ca^{2+}]_i$ increase in the flagellum

It is widely accepted that an increase in $[Ca^{2+}]_i$ precedes AE (*Sánchez-Cárdenas et al., 2014*; *Romarowski et al., 2016*). It has been shown that AE initiation by progesterone occurs after an increase in $[Ca^{2+}]_i$ in the sperm head that is later propagated toward the midpiece (*Romarowski et al., 2016*). To comprehend the connection between AE and the concomitant contraction of the midpiece, we hypothesized the existence of a signal, that is, $Ca^{2+}$, which propagates from the head to the flagellum and modulate the architecture of the sperm flagellum.

To investigate if the increase in the concentration of $[Ca^{2+}]_i$ in the midpiece is correlated with the midpiece contraction, sperm were incubated with FM4-64 and Fluo4, a $Ca^{2+}$ fluorescent sensor (*Figure 4—video 1*). *Figure 4A* shows a live-cell imaging experiment of sperm cells that did not undergo AE (control case), nor a significant rise in $[Ca^{2+}]_i$. Noteworthy, upon induction with progesterone (*Figure 4C*, *Figure 4—video 1*), there was an increase in the midpiece $[Ca^{2+}]_i$ followed by an increase in FM4-64 fluorescence, which is indicative of a contraction of the flagellum (see *Figure 2—figure supplement 1D*). These single-cell experiments illustrate a reduction of the midpiece diameter that occurs concomitantly with an increase in $[Ca^{2+}]_i$ within the flagellum. A large population of sperm was analyzed using low magnification (×10). *Figure 4B and D* show a collection of single-cell kymographs, where each row represents the fluorescence of Fluo4 (green) and FM4-64 (red) of the midpiece over time. Unstimulated cells are displayed on *Figure 4B*, and sperm that were exposed to progesterone are shown in *Figure 4D*. Most of the cells display a transient $[Ca^{2+}]_i$ increase followed by an increase in FM4-64 fluorescence. This pattern was also observed in ionomycin-induced cells (*Figure 4—figure supplement 1A and B* and *Figure 4—video 1*) with the exception that, as expected, in this case the rise in $[Ca^{2+}]_i$ was faster and sustained. These findings indicate that an $[Ca^{2+}]_i$ transient increase, happening in the midpiece, precedes the contraction of the flagellum.

The relationship between the observed $[Ca^{2+}]_i$ rise and the flagellar contraction was then assessed using live-cell super-resolution imaging. *Figure 4E* shows there is an $[Ca^{2+}]_i$ increase in the midpiece that coincides in space and time with a contraction of the midpiece. To provide insight in the spatiotemporal relationship of the $[Ca^{2+}]_i$ rise and the midpiece contraction, these results were visualized using a 3D kymograph encompassing the dynamics happening along the entire midpiece (*Figure 4F*). Overall, a rise in $[Ca^{2+}]_i$ occurred along the whole midpiece, which coincided with a generalized contraction of the midpiece (*Figure 4F*). Remarkably, a transient focal increase of $[Ca^{2+}]_i$ was observed at the base of the flagellum, which correlates in space and time with a focal reduction of the diameter (*Figure 4F*). We then sought to identify a molecular/structural link between both processes.

## The distance between the actin cytosk (geleton and the plasma membrane is decreased during the contraction of the midpiece

The midpiece is shaped by two major structural elements: the axoneme, which consists of tubulin and its accessory components, including the ODFs; and a recently described network of filamentous actin (F-actin) arranged in a helicoidal conformation along the midpiece (*Gervasi et al., 2018*). We investigated whether the actin network plays a role in regulating the midpiece contraction associated with AE. We considered three possible scenarios: (1) an unknown mechanical force, mediated by a rigid matrix such as F-actin, brings the plasma membrane closer to the center of the flagellum; (2) a dynamic relationship exists between the plasma membrane and the actin cytoskeleton of the midpiece, facilitating contraction; and (3) the midpiece contraction is driven by an uncharacterized signaling mechanism that acts independently of the actin cytoskeleton.

The role of the actin cytoskeleton in the midpiece contraction was investigated through live-cell super-resolution using SiR-actin, a fluorescent probe that binds to F-actin (*Lukinavičius et al., 2014*; *Magliocca et al., 2017*; *Yamazaki et al., 2018*). We used FM4-64 to visualize the plasma membrane (*Figure 5*) and transgenic sperm expressing DsRed2 to observe the mitochondrial network

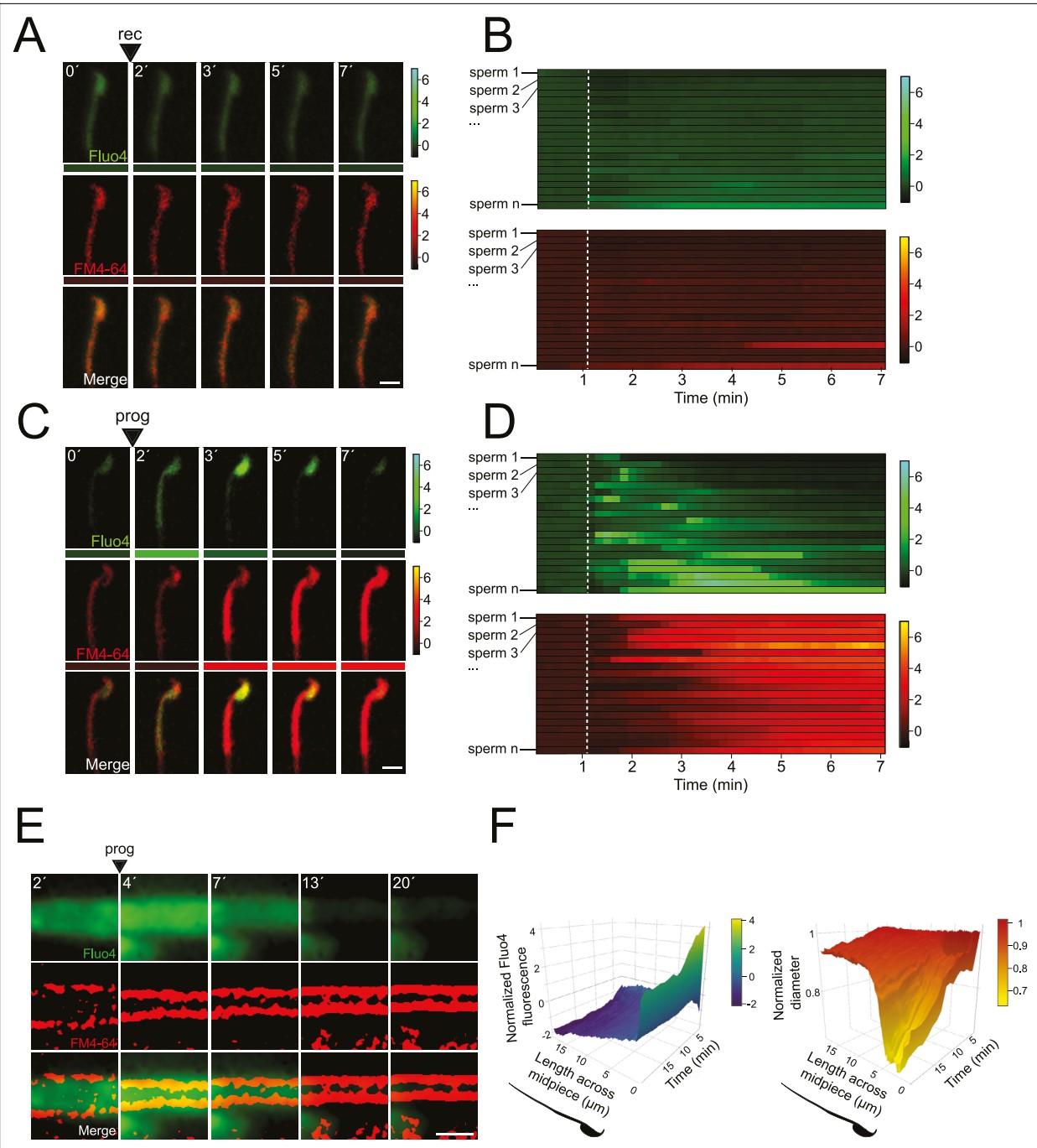

**Figure 4.** Midpiece contraction is driven by $[Ca^{2+}]_i$ changes. (**A**) The representative time series demonstrates $[Ca^{2+}]_i$ and acrosomal exocytosis (AE) dynamics. Capacitated F1 sperm, loaded with Fluo4 AM, were immobilized on concanavalin A-coated coverslips and incubated in a recording medium (rec) containing 10 μM FM4-64. Rec was added as indicated by arrowheads. Scale bar = 10 μm. Beneath each frame in the Fluo4 (green) and FM4-64 (red) images, a color code displays the normalized intensity of the fluorescence signal (scale bar on the right of the panel). (**B**) Kymograph-like analysis of the midpiece of 20 sperm following the addition of recording medium. Each row depicting the $[Ca^{2+}]_i$ (upper) and membrane (lower) dynamics of a single cell over time. A white dotted line indicates the moment of addition. The images presented are representative of at least five independent experiments. (**C**) The representative time series demonstrates $[Ca^{2+}]_i$ and AE dynamics. Progesterone (prog, 100 μM) was added as indicated by arrowheads. Scale bar = 10 μm. Beneath each frame in the Fluo4 (green) and FM4-64 (red) images, a color code displays the normalized intensity of the fluorescence signal (scale bar on the right of the panel). (**D**) Kymograph-like analysis of the midpiece of 20 sperm following the addition of prog. Each row depicting the $[Ca^{2+}]_i$ (upper) and membrane (lower) dynamics of a single cell over time. A white dotted line indicates the moment of addition. The images presented are representative of at least five independent experiments. Consistently, an $[Ca^{2+}]_i$ transient increase precedes contraction, which is proportional to the increase in FM4-64 fluorescence, as shown in *Figure 2—figure supplement 1D*. (**E**) Representative time series of $[Ca^{2+}]_i$ and

*Figure 4 continued on next page*

*Figure 4 continued*

midpiece contraction dynamics. Capacitated CD1 sperm were loaded with Fluo4 AM, immobilized on concanavalin A-coated coverslips, and incubated in a recording medium containing 0.5 µM FM4-64. AE was induced with 100 µM progesterone (prog, arrowhead). Fluo4 images are widefield images, while FM4-64 images are super-resolution radial fluctuations (SRRF)-processed (super-resolution). Scale bar = 1 µm. (**F**) 3D kymographs of $[Ca^{2+}]_i$ (left) and contraction (right) dynamics. Data are normalized to the mean of the frames before the induction of AE. Representative images from at least five independent experiments are shown, with 36 cells analyzed.

The online version of this article includes the following video and figure supplement(s) for figure 4:

**Figure supplement 1.** $[Ca^{2+}]_i$ concentration drives midpiece contraction in Ionomycin-stimulated sperm.

**Figure 4—video 1.** Representative movies showing $[Ca^{2+}]_i$ and acrosomal exocytosis (AE) dynamics for a control case (addition of recording medium, rec), progesterone (Prog, 100 µM), and ionomycin-induced AE (Iono, 10 µM).

https://elifesciences.org/articles/93792/figures#fig4video1

(*Figure 5—figure supplement 1*). Two stimulated sperm, one that experienced AE (*Figure 5B*) and another that did not (*Figure 5A*) are shown. In both cases, a network of F-actin was observed beneath the plasma membrane of the midpiece (cortical actin), enveloping the mitochondrial network (*Figure 5—figure supplement 1A*). As expected, in the absence of AE, the midpiece diameter (observed through FM4-64) remained unchanged (see also *Figure 2* and *Figure 5—figure supplements 1–4*). In this case, both the diameter of the F-actin network and its proximity to either the plasma membrane or the mitochondrial network remained static (*Figure 5A*), suggesting a structural role for the F-actin network within the midpiece [scenario (1)], such as supporting the organization of the mitochondrial network (*Figure 5—figure supplements 1–4*). Remarkably, acrosome-reacted sperm (*Figure 5B*) experienced an abrupt structural reorganization of the midpiece, characterized by a decrease in the distance between the plasma membrane and both the F-actin and mitochondrial networks (*Figure 5—figure supplements 1 and 2*). This observation suggests that the remodeling of the midpiece structure is driven by a mechanical (or molecular) interaction occurring at the boundaries of the plasma membrane [supporting scenario (2)].

To assess the potential interaction between the F-actin network and the plasma membrane (or its associated components), we examined their colocalization through live-cell super-resolution imaging. *Figure 5C* presents two Manders' colocalization coefficients (*Dunn et al., 2011*). M1 represents the proportion of F-actin (in pixels) that colocalizes with the plasma membrane, while M2 measures the proportion of plasma membrane pixels colocalizing with the F-actin network. A Manders' value of 1 indicates full colocalization, while a value of 0 implies no colocalization.

*Figure 5C* reveals that acrosome-intact sperm displayed low Manders' coefficients (M1 = 0.32 ± 0.03; M2 = 0.28 ± 0.04). In contrast, sperm undergoing midpiece contraction exhibited a significant increase in colocalization between the F-actin network and the plasma membrane, as evidenced by the rise in M1 and M2 coefficients (M1 = 0.65 ± 0.03; M2 = 0.69 ± 0.03). This increased colocalization occurred simultaneously with midpiece contraction.

Overall, these results confirm that (1) the contraction of the midpiece is linked to the remodeling of the plasma membrane and potentially the F-actin cytoskeleton, and that (2) both structures interact with each other, directly or indirectly, at the nanoscales. We then seek to understand whether the driving force for the contraction of the midpiece emanates from the F-actin network. The following scenarios were envisaged: (1) the plasma membrane moves toward the actin cytoskeleton, (2) the actin cytoskeleton moves toward the plasma membrane, or (3) both structures come closer to the center of the flagellum. To investigate the occurrence of any of these scenarios, the positions of fluorescence peaks of FM4-64 and SiR-actin in super-resolution kymographs were tracked, as proxy of either membrane or actin cytoskeleton localization, respectively (*Figure 5—figure supplement 2A–F*). *Figure 5—figure supplement 2D–F* show representative histograms of SiR-actin and FM4-64 over time for ionomycin, progesterone-induced and spontaneous AE, respectively. Both FM4-64 and SiR-actin fluorescence peaks came closer to each other. This effect can be also seen in *Figure 5—figure supplement 2I*, where acrosome-intact cells display a distance of 0.180 ± 0.007 µm between the plasma membrane and the actin cytoskeleton. In acrosome-reacted cells, the distance was as small as 0.074 ± 0.006 µm after midpiece contraction.

In *Figure 5—figure supplement 2D–G*, a section of the midpiece of an ionomycin-induced AE sperm is shown. In this case, the plasma membrane and the actin cytoskeleton move closer toward the center of the cell (set to 0). The slope calculated from the linear fit denotes the velocity of this change.

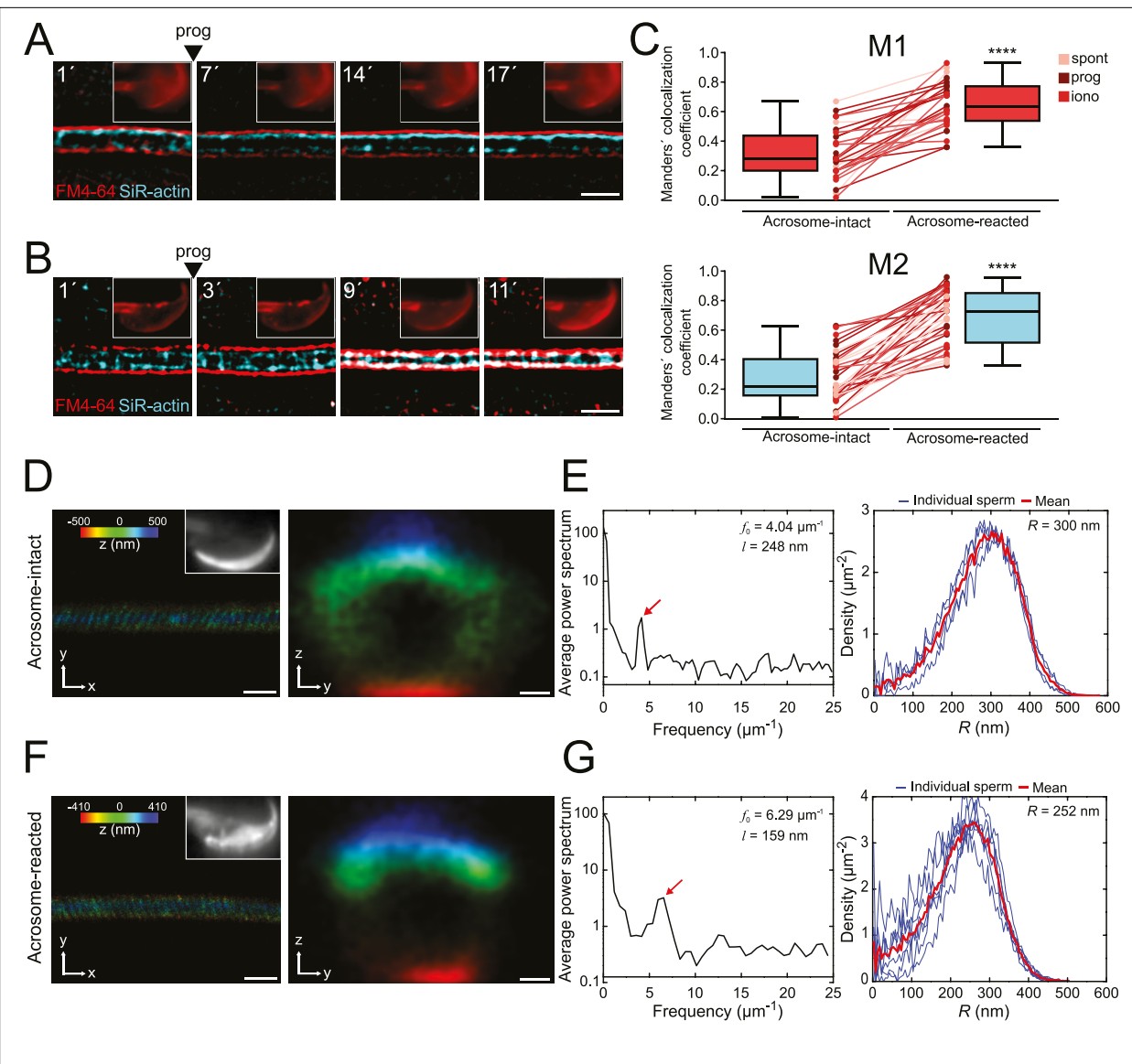

**Figure 5.** The flagellar membrane approaches the actin cytoskeleton in the midpiece of the sperm flagellum during midpiece contraction and acrosomal exocytosis (AE). (**A, B**) Representative time series of plasma membrane and actin cytoskeleton colocalization in the midpiece in the absence of AE (**A**) and during progesterone-induced AE (**B**, prog, 100 μM; FM4-64 shown in red, SiR-actin shown in cyan). Capacitated CD1 sperm were loaded with 100 nM SiR-actin, immobilized on concanavalin A-coated coverslips, and incubated in a recording medium containing 0.5 μM FM4-64. Scale bar = 1 μm. Representative images from at least five independent experiments are shown, with 36 cells analyzed. (**C**) Manders' colocalization coefficients for acrosome-intact and acrosome-reacted cells in the midpiece. M1 was assigned to FM4-64, and M2 to SiR-actin. Data are presented as mean ± SEM. ****p<0.0001 represents statistical significance. Paired *t*-test was performed. (**D–G**) Representative images of sperm midpiece stained with the acrosome marker PNA (left panel, upper right insets, epifluorescence) and phalloidin (actin filaments, STORM) for acrosome-intact (**D, E**) and acrosome-reacted (**F, G**) cells. The left panel displays a longitudinal section of the midpiece (scale bar = 10 μm), while the right panel illustrates the radial distribution (scale bar = 0.2 μm). (**E, G**) Schematics of the analyzed actin double helix parameters in the midpiece: helical pitch (*l*, distance between turns of the helix, left panel), helical pitch frequency (*f₀*, number of turns the helix makes per 1 μm), and radial distribution (*R*, radius of the double helix, right panel). Representative images from at least three independent experiments are shown. Four acrosome-intact cells and seven acrosome-reacted cells were analyzed.

The online version of this article includes the following figure supplement(s) for figure 5:

**Figure supplement 1.** Unaltered distance between F-actin and mitochondrial network after acrosomal exocytosis (AE) indicates consistent mitochondrial network organization in EGFP-DsRed2 transgenic sperm.

**Figure supplement 2.** Plasma membrane approaches actin cytoskeleton during midpiece contraction.

**Figure supplement 3.** Actin cytoskeleton reorganization in the midpiece during acrosomal exocytosis (AE).

**Figure supplement 4.** Midpiece contraction occurs following sperm-egg fusion.

In this example, the plasma membrane is contracting at a rate of 14 μm/min, whereas the actin cytoskeleton is contracting at a rate of 3 μm/min (*Figure 5—figure supplement 2E and H*). In both cases, the signal presents a negative slope in sperm that underwent AE, consistent with a decrease in midpiece diameter. Since the results of the analysis of SiR-actin slopes were not conclusive, we studied the actin cytoskeleton structure in more detail.

## The actin cytoskeleton in the midpiece is remodeled during the AE

In the midpiece, polymerized actin forms a double helix that accompanies mitochondria (*Gervasi et al., 2018*). To investigate if the contraction of the midpiece is associated with structural modifications of the actin double-helix, 3D-STORM was used. Cells were exposed to progesterone, fixed, and co-stained with PNA and phalloidin to visualize the acrosomal status and the actin structure, respectively.

*Figure 5D and F* show representative images of both dyes for acrosome-intact and acrosome-reacted sperm (left panels). Different parameters were calculated: (1) Helical pitch (*l*), which is the distance between turns of the helix; (2) frequency ($f_0 = 1/l$), obtained from the Fourier transform of the image and represented the number of turns that the helix makes per unit length; and (3) radial distribution (*R*), to infer the distance between the center of the midpiece and the maximum of fluorescence (*Gervasi et al., 2018*). The frequency increased substantially in the acrosome-reacted sperm compared to intact sperm (6.29 vs. 4.04 μm$^{-1}$, *Figure 5E and G*, red arrows in left panels). In addition, the helical pitch diminished its magnitude in acrosome-reacted sperm (159 nm vs. 248 nm) (*Figure 5E and G*, left panels). The radial distribution of F-actin in cells that underwent AE indicated a smaller radius compared to acrosome-intact sperm (252 vs. 300 nm, *Figure 5E and G*, right panel). Altogether, these results confirm that the actin double helix undergoes structural changes during the contraction of the midpiece.

To further investigate the structural rearrangements of the actin cytoskeleton during AE, a fluorescent molecule number and brightness analysis was performed (*Digman et al., 2008*; *Mandracchia et al., 2020*; *Liu et al., 2017*). The number measure indicates the abundance or concentration of SiR-actin molecules bound to F-actin fibers, while the brightness analysis helps reveal dynamic changes in molecular aggregation processes. This analysis is based on the idea that higher-order fluorescent complexes, which are mobile within the sample, will cause an increase in signal variability over time. An increase in brightness suggests the formation or movement of supramolecular structures, such as bundles of actin fibers bearing SiR-actin. AE induction with progesterone and ionomycin resulted in a significant increase in actin number and brightness compared to the control (non-reacted) group (*Figure 5—figure supplement 4A*). Spontaneously reacted sperm also exhibited this behavior, albeit in a more moderate fashion (*Figure 5—figure supplement 4A*).

The observed increase in SiR-actin number after AE induction suggests major actin monomer recruitment to the polymerizing fibers (*Figure 5—figure supplement 4A and B*), which produces an apparent increase in local fluorophore concentration (*Figure 5—figure supplement 4B*). This effect positively correlated with a similar increase in SiR-actin fluorescence (*Figure 5—figure supplement 4A*), which is known to indicate actin polymerization due to its high affinity for F-actin.

Actin cytoskeleton polymerization in the midpiece results in a signal variation of increased magnitude (an increase in SiR-actin brightness) as it undergoes structural rearrangements (*Figure 5—figure supplement 4A and B*), which are compatible with other observations reported in this work. Both assembly and transport of actin complexes into higher oligomeric states (F-actin) across the cellular milieu led to an apparent increase in the registered SiR-actin brightness (*Figure 5—figure supplement 4B*). Additionally, actin filament displacement along the mitochondrial sheath during AE provides a SiR-actin brightness measure with high dispersion, indicating a redistribution of this cellular structure. These results suggest that actin filaments are locally redistributed and remodeled during AE (*Figure 5—figure supplement 4B*).

## Midpiece contraction in sperm located within the perivitelline space

In mice, only sperm undergoing AE prior to binding to the zona pellucida can penetrate and fertilize (*Jin et al., 2011*). We hypothesized that midpiece changes and motility cessation occur only after acrosome-reacted sperm penetrate the zona pellucida. Live imaging was performed after in vitro

fertilization (IVF) using transgenic EGFP-DsRed2 sperm loaded with FM4-64. Eggs were inseminated with a high sperm count (200,000 cells) to increase the number of cells observed.

*Figure 6A* and *Figure 6—video 1* show a sperm swimming within the perivitelline space. The FM4-64 fluorescence of this sperm midpiece is low (*Figure 6B*), which coincides with the fact that the sperm is moving (see also *Figure 1F and G*). Another example of this observation is shown in *Figure 6C and D*, an acrosome-reacted moving sperm within the perivitelline space had low FM4-64 fluorescence in the midpiece (*Figure 6C*). After 20 min, these sperm stopped moving and exhibited increased FM4-64 fluorescence, indicating midpiece contraction (*Figure 6D*). These results suggest that midpiece contraction and motility cessation occur after acrosome-reacted sperm penetrate the zona pellucida.

## Midpiece contraction takes place following sperm-egg fusion

Motile acrosome-reacted sperm, initially lacking midpiece contraction, can penetrate the zona pellucida. However, they eventually stop moving and exhibit an increase in FM4-64 fluorescence. This suggests that the cessation of motility plays a crucial role in fertilization events occurring after zona pellucida binding and penetration, and that a similar change in midpiece architecture is necessary for successful sperm-egg fusion. To explore this hypothesis, we designed a live-imaging experiment using zona-free eggs. Denuded oocytes were loaded with Hoechst 33342, a nuclear dye, to visualize the exact moment of sperm-egg fusion (*Conover and Gwatkin, 1988*; *Hinkley et al., 1986*; *Stewart-Savage and Bavister, 1988*). The experiment involved FM4-64, transgenic EGFP-DsRed2 sperm, and simultaneous signal collection from five separate channels (DIC, Hoechst, EGFP, DsRed2, and FM4-64). To enhance observation likelihood, volumetric data was collected by imaging at different z-planes (*Figure 7A*).

*Figure 7C* presents a representative volumetric time series of sperm-egg fusion, with only the optimal focal plane shown. The acrosome-reacted sperm (EGFP negative) initially did not display Hoechst fluorescence (*Video 1*), indicating a lack of fusion. However, 14 min later, the dye entered the sperm and stained the nucleus, indicating fusion initiation. At this stage, sperm remain motile while bound to the egg plasma membrane. The midpiece diameter remained unchanged during initial sperm-egg fusion indicated by the low FM4-64 fluorescence. Following gamete fusion initiation, FM4-64 fluorescence increased (38–44 min), causing sperm motility to cease. In some instances, the midpiece folded and extended again (*Figure 7—figure supplement 1B*). These findings demonstrate that midpiece contraction occurs following sperm-egg fusion.

## A decrease in [Ca$^{2+}$]$_i$ in the midpiece following fusion precedes the midpiece contraction

In previous experiments, an increase in [Ca$^{2+}$]$_i$ was observed throughout the midpiece, coinciding with midpiece contraction (*Figure 4*, *Figure 4—figure supplement 1*). To examine the mechanism behind midpiece contraction after fusion, wild-type sperm loaded with Fluo4 were exposed to denuded oocytes loaded with Hoechst. We hypothesized that dynamic changes in [Ca$^{2+}$]$_i$ during gamete fusion drive the midpiece contraction during sperm immobilization.

*Figure 8A* presents representative images from a time-course live-imaging experiment designed to observe sperm-egg fusion. The experiment involved simultaneous signal collection from four separate channels over time (DIC, Hoechst, Fluo4, and FM4-64). To enhance observation likelihood, images were taken at different z-planes, though only the optimal focal plane is shown.

Unexpectedly, sperm binding to the egg plasma membrane displayed high levels of [Ca$^{2+}$]$_i$ in the head and midpiece (*Figure 8A*, time 1 min, and *Video 2*). A few minutes after fusion began (indicated by an increase in Hoechst signal in the sperm), a decrease in [Ca$^{2+}$]$_i$ in the head and midpiece was observed (21–51 min). This decrease in [Ca$^{2+}$]$_i$ was followed by midpiece contraction, as evidenced by the increase in FM4-64 fluorescence (56 min). In several cells, the midpiece folded back before contracting. The gametes that underwent this change in the flagellum typically remained folded, but occasionally the tail unfolded again. *Figure 8B–F* quantify the key events described above during sperm-egg fusion.

In summary, 75.88 ± 12.20% of the sperm exhibited midpiece contraction upon fusion. In 61.83 ± 12.04% of cases, the midpiece folded on itself, and afterward, only 31.60 ± 7.47% unfolded and stretched out again. All sperm bound to the plasma membrane before fusion presented high [Ca$^{2+}$]i in

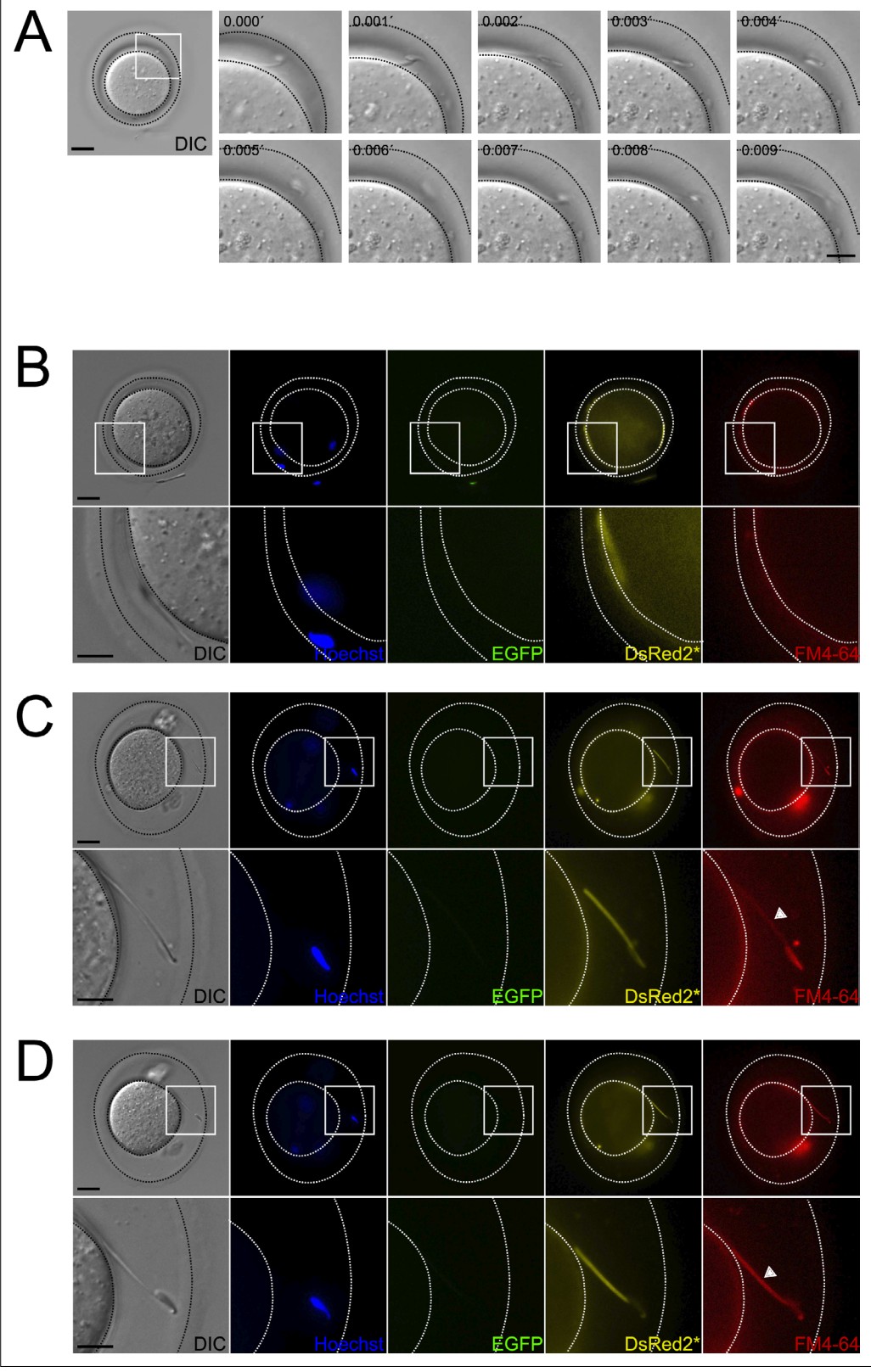

**Figure 6.** Occurrence of midpiece contraction in sperm located within the perivitelline space. Representative images of in vitro fertilization (IVF) experiments using EGFP-DsRed2 sperm. Oocyte-sperm complexes were stained with 10 μg/ml Hoechst and 10 μM FM4-64. (**A**) Representative time series of DIC images showing a sperm moving within the perivitelline space. Scale bar in right panel = 20 μm, scale bar in left panel = 10 μm. (**B**) DIC, Hoechst,

*Figure 6 continued on next page*

*Figure 6 continued*

EGFP, DsRed2*, and FM4-64 images are shown for the case depicted in (**A**), note that, as the sperm is moving, it is located in a different position in the perivitelline space. The area depicted in the upper panel is shown in higher magnification in the lower panel. Scale bar in upper panel = 20 μm, scale bar in lower panel = 10 μm. (**C, D**) DIC, Hoechst, EGFP, DsRed2*, and FM4-64 images are shown for a (**C**) sperm that have passed through the ZP, displaying acrosomal exocytosis (AE) with an initially non-contracted midpiece. After 20 min, as shown in (**D**), the midpiece becomes contracted. The area depicted in the upper panel is shown in higher magnification in the lower panel. Scale bar in upper panel = 20 μm, scale bar in lower panel = 10 μm. Representative images from at least six independent experiments are shown. A total of 23 oocytes and 69 sperm were analyzed.

The online version of this article includes the following video for figure 6:

**Figure 6—video 1.** Representative movie of in vitro fertilization (IVF) assay.
https://elifesciences.org/articles/93792/figures#fig6video1

---

the head, and the majority displayed a decrease in head $[Ca^{2+}]_i$ before midpiece contraction (75.13 ± 3.05%). Concerning the midpiece, three patterns were observed during fusion: in the majority of cases (69.67 ± 11.46%), the midpiece experienced a decrease in $[Ca^{2+}]_i$. Additionally, 10.33 ± 10.33% of sperm showed a transient increase in $[Ca^{2+}]_i$, while the remaining 20.00 ± 9.41% displayed no changes.

Collectively, these results indicate that a decrease in $[Ca^{2+}]_i$ in the midpiece after fusion precedes midpiece contraction and the cessation of sperm motility that precedes sperm-egg fusion.

## Discussion

In this article, we demonstrate the existence of a fundamental structural change that occurs in the sperm flagellum at the time of fusion with the egg. Using a plethora of advanced microscopy methods and single-cell imaging, we provide insight about cellular and molecular events that occur in acrosome-reacted sperm, which, undoubtedly, are the ones capable of fertilizing an oocyte (*Hino et al., 2016*; *La Spina et al., 2016*). At this precise moment of the fertilization process, sperm need to stop moving to complete the fusion between the two gametes. The cease of movement is caused by two concomitant processes that take place in the flagellar midpiece region: a change in the F-actin helical structure and a decrease in the midpiece diameter. To the best of our knowledge, this is the first time that a structural modification of the sperm flagellum related to a specific cellular necessity, that is, gamete fusion, is described.

To arrive at the site of fertilization within the female reproductive tract, sperm motility is required. It is also fundamental to penetrate the different layers surrounding the egg. In this journey, sperm sense the environment and adapt their movement to the different physiological scenarios. In addition, recent evidence also demonstrated that after being released from their site of storage in the oviductal isthmus, mouse sperm undergo AE (*Hino et al., 2016*; *La Spina et al., 2016*). This process occurs in the upper segments of the oviduct before any interaction with the eggs or their surrounding layers. This observation opened a new scenario about the motility of sperm in their last transit to the site of fertilization. Little is known about sperm in that region and how sperm move after AE. In this regard, most of the information about mammalian sperm comes from studies conducted in vitro, using a mixture of acrosome-intact and acrosome-reacted sperm. Even if their motility is analyzed in a subjective manner or using cell tracking systems, virtually all the experiments are conducted without discerning the acrosomal status of the cells. Another possible source of artifacts in this analysis is related with the fact that most of the experiments are also performed using epididymal sperm (not ejaculated) in aqueous solutions that support sperm capacitation but do not represent the natural viscous environment present in the female tract.

By performing in vitro experiments, we detected a strong association between the decrease in the midpiece diameter and the cease of sperm motility. This is observed in a subset of sperm after the occurrence of AE. This may suggest that only sperm that undergo AE in the oviduct and do not experience this midpiece contraction are capable of migrating to the ampulla and penetrate the cumulus and the zona pellucida. Those sperm that undergo this phenomenon earlier in the tract may not be suitable to continue their journey, suggesting that this also may select the gametes during their transit to the ampulla. Previous observations tracking sperm within the female tract have shown the existence

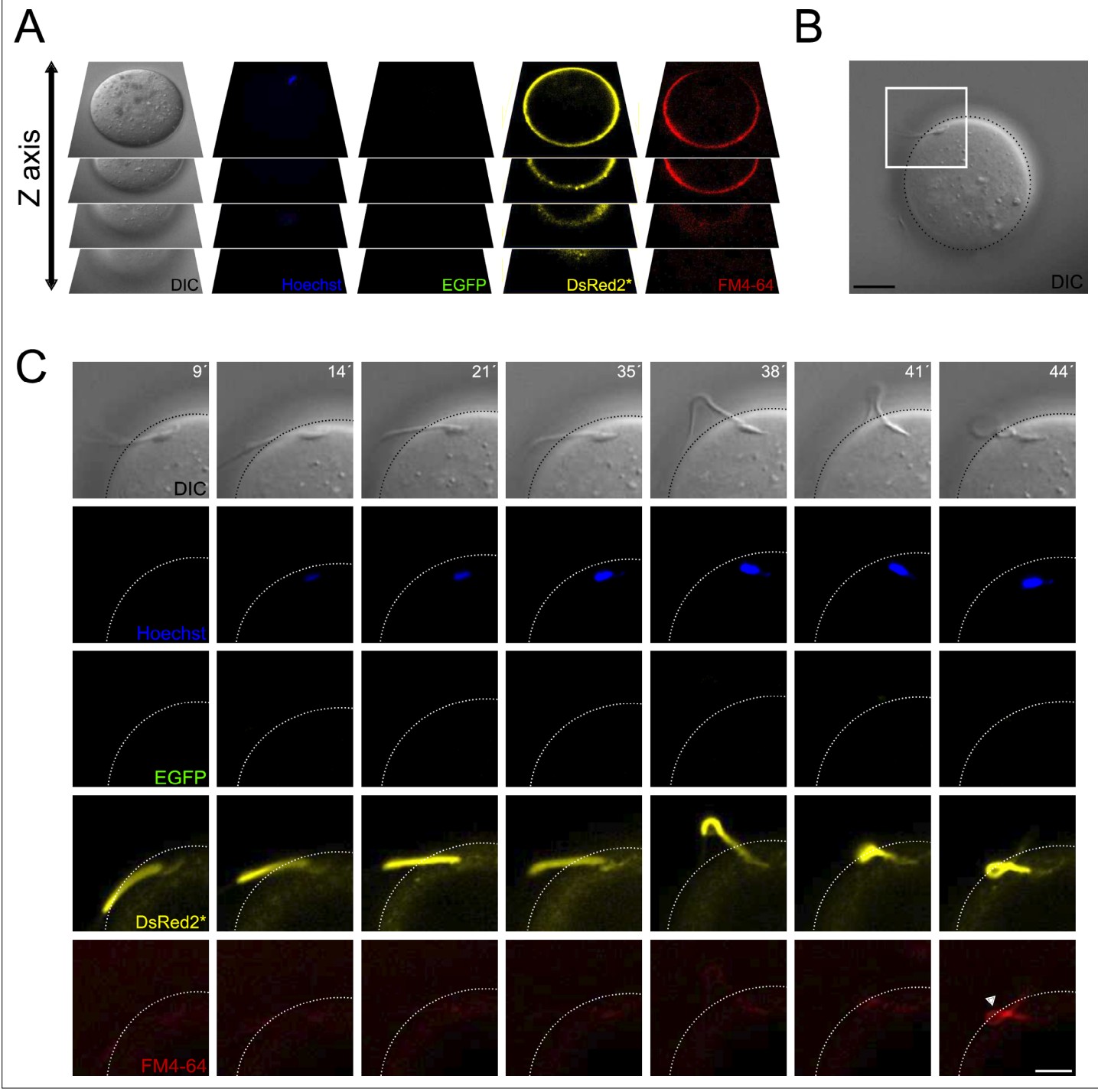

**Figure 7.** Contraction of the midpiece occurs after sperm-egg fusion. (**A**) Schematic representation of the acquisition settings for the sperm-oocyte fusion assay. Images were taken every 7 μm along the z-axis. (**B**) DIC image of a sperm-oocyte complex, with the area depicted in higher magnification in panel (**C**). Scale bar = 20 μm. (**C**) Representative time series of sperm-oocyte fusion assay experiments using EGFP-DsRed2 sperm. Oocytes were stained with 1 μg/ml Hoechst and 10 μM FM4-64. DIC, Hoechst, EGFP, DsRed2*, and FM4-64 images are shown over time. Scale bar = 10 μm. Note that midpiece contraction occurs after sperm-egg fusion and is proportional to the increase in FM4-64 fluorescence, as shown in *Figure 2—figure supplement 1D*, highlighting its potential importance in the fertilization process. Representative images from at least four independent experiments are displayed.

The online version of this article includes the following figure supplement(s) for figure 7:

**Figure supplement 1.** Moment-based analysis overview.

**Video 1.** Representative movie of sperm-oocyte fusion assay experiments using EGFP-DsRed2 sperm. Oocytes were stained with 1 µg/ml Hoechst and 10 µM FM4-64. DIC images are presented in gray scales, Hoechst in blue, EGFP is green, DsRed2* in yellow, and FM4-64 in red. Sperm of interest is indicated with a white arrow. https://elifesciences.org/articles/93792/figures#video1

of acrosome-reacted sperm within the tract that remain non-motile (*Hino et al., 2016*; *La Spina et al., 2016*; *Muro et al., 2016*).

Acrosome-reacted sperm that bind and penetrate the zona pellucida are ready to fuse with the egg. During sperm-egg fusion, several authors have reported in different species that sperm stop moving (*Gaddum-Rosse et al., 1984*; *Gaddum-Rosse et al., 1982*; *Ravaux et al., 2016*). However, the mechanism behind this behavior is not established. One possible explanation is that fusion promotes ion transport changes in sperm, which significantly alter flagellar movement. In this sense, it was previously demonstrated that certain ions such as $Ca^{2+}$ may diffuse from the oocyte to the sperm (*Jones et al., 1998*). However, a clear technical limitation in our experimental approach

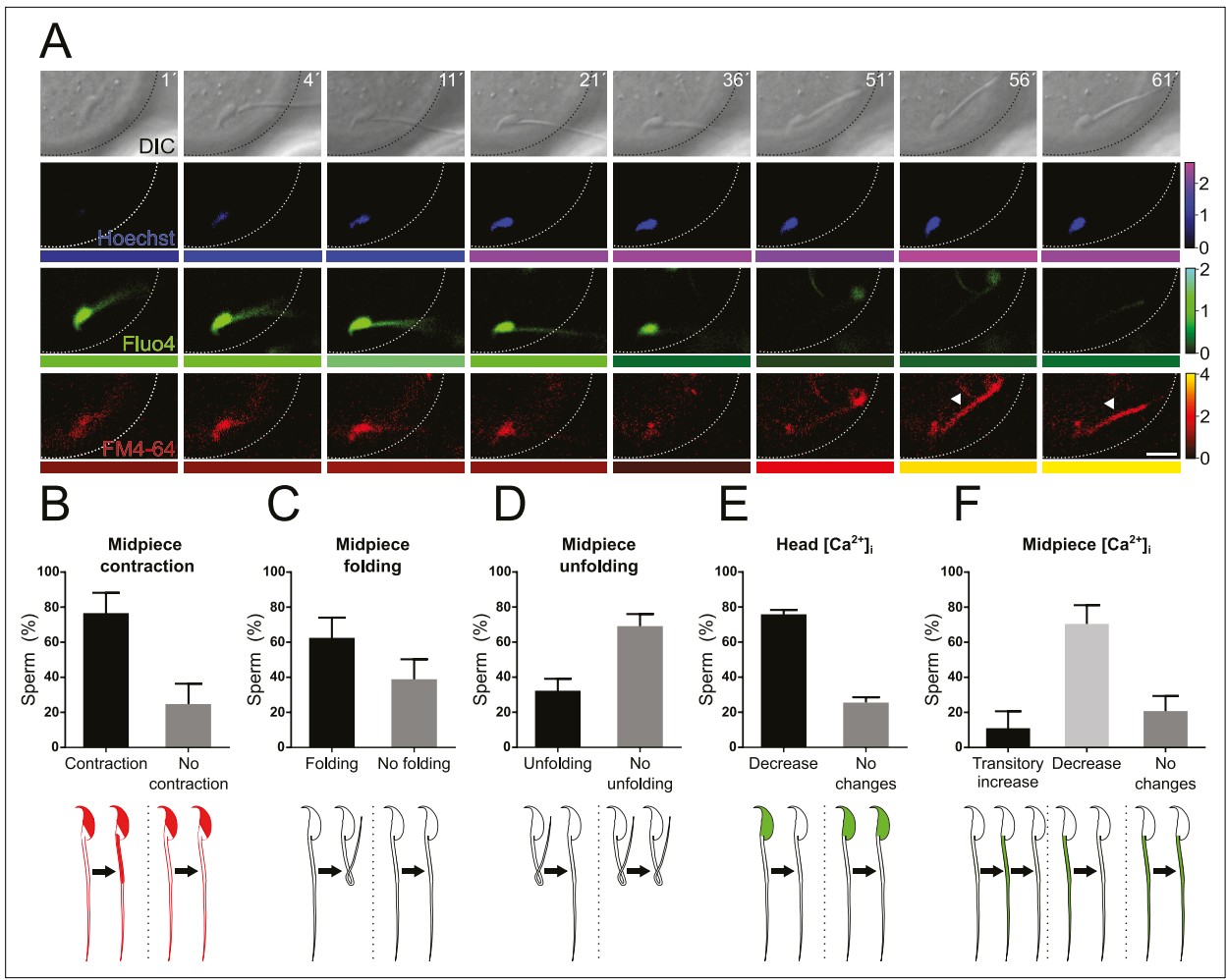

**Figure 8.** Midpiece contraction occurs in sperm-egg fusion after a decrease in $[Ca^{2+}]_i$. (**A**) Representative time series of sperm-oocyte fusion assay experiments using wild-type sperm loaded with 1 µM Fluo-4. Oocytes were stained with 1 µg/ml Hoechst and 10 µM FM4-64. DIC, Hoechst, Fluo-4, and FM4-64 images are shown over time. Scale bar = 10 µm. The color code below each frame in the Hoechst (shown in blue), Fluo-4 (shown in green), and FM4-64 (shown in red) images indicates the normalized intensity of the fluorescence signal (scale bar on the right of the panel). (**B–F**) Quantification of sperm showing midpiece contraction (**B**, indicated by increased FM4-64 fluorescence), midpiece folding (**C**), midpiece unfolding (**D**), Fluo-4 fluorescence dynamics in the head (**E**), and different patterns in the midpiece (**F**) during fusion. Data are presented as the mean ± SEM of the percentage of sperm counted for each experiment. Representative images and data from at least three independent experiments are shown. A total of 74 oocytes and 136 sperm were analyzed. Note that midpiece contraction occurs in sperm-egg fusion after a decrease in Fluo-4 fluorescence.

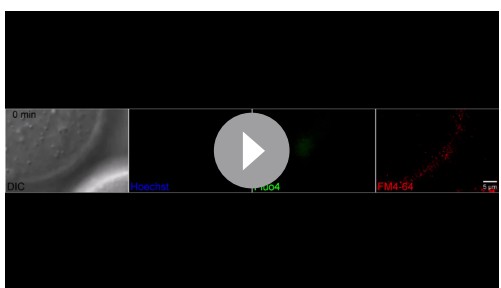

**Video 2.** Representative movie of sperm-oocyte fusion assay experiments using wild-type sperm loaded with 1 µM Fluo-4. Oocytes were stained with 1 µg/ml Hoechst and 10 µM FM4-64. DIC images are presented in gray scales, Hoechst in blue, Fluo4 in green, and FM4-64 in red.

https://elifesciences.org/articles/93792/figures#video2

is that the probes that monitor ion dynamics may be exchanged between both gametes. If the concentration of a given probe does not remain stable, it is impossible to determine the accurate change that occurs during fusion. Future experiments may take advantage of transgenic models that incorporate a particular sensor to study this process in vivo, such as the one used by *Cohen et al., 2022*.

Regardless of the precise nature of ion exchange between sperm and eggs, the diameter of the sperm flagellum in the midpiece is reduced. This midpiece diameter reduction is strongly associated with cessation of sperm motility and is apparently needed to complete the fusion process. Our observation clearly supports this notion. Importantly, the sperm flagellum folds back during fusion coincident with the decrease in the midpiece diameter. Interestingly, this was previously observed in mammals as well as in sea urchin sperm (*Gaddum-Rosse et al., 1984*; *Darszon et al., 2008*; *Guerrero et al., 2010*).

Like other cylindrical biological structures, the sperm flagellum relies on the cytoskeleton for its structural organization and specialized mechanical properties. In addition to the change in midpiece diameter, a significant rearrangement of the F-actin cytoskeleton also occurs. In the midpiece, the polymerized actin is organized in a double helix accompanying the mitochondria (*Gervasi et al., 2018*). As in many other organisms, the actin cytoskeleton possesses important structural functions, and dynamic changes of F-actin allow the cells to conduct important cellular tasks such as exocytosis. However, less is known about the structural changes undergone by the actin cytoskeletons in cilia and flagella. Remarkably, it has been reported that actin forms helix-like structures in the flagellum of the parasite *Giardia intestinalis* (*Paredez et al., 2011*). These findings open the question of whether, to some extent, flagellar helical structures are conserved among diverse species. Our observations demonstrate that the F-actin double helix undergoes a change in the helical pitch as well as in the radial distance to the axoneme. This is concomitant with the decrease in midpiece diameter. Our single-cell experiments using super-resolution microscopy also revealed that the plasma membrane approached the F-actin network during this change. It is well established that various proteins can function as linkers between the plasma membrane and the actin cytoskeleton (*Köster and Mayor, 2016*), but their roles in this specific process remain to be studied. Regardless of how these structures are connected, it is evident that both are associated. However, our experimental data cannot determine whether the plasma membrane is causing the change of the actin network or if the actin network influences the plasma membrane.

Another observation emerging from our study is that a change in $[Ca^{2+}]_i$ occurs prior to the midpiece contraction. It is well known that $Ca^{2+}$ is important for AE, and a specific transient rise in $[Ca^{2+}]_i$ originating in the head can trigger exocytosis (*Romarowski et al., 2016*). Previous observations have demonstrated that the sperm head and tail are not isolated compartments, and that ions and other molecules can move between them (*Buffone et al., 2012*; *De La Vega-Beltran et al., 2012*). In our single-cell experiments, we observed that the rise in FM4-64 fluorescence in the midpiece occurs after the increase in $[Ca^{2+}]_i$ in that region, suggesting a potential association between these events. This could involve diffusion or active transport processes; further investigation is required to determine the precise mechanism to demonstrate if the structural changes are triggered by $Ca^{2+}$. In addition, $[Ca^{2+}]_i$ increase and/or modification in the midpiece architecture may result in functional changes in the mitochondria such as the status of the mitochondrial membrane potential and the ATP production. This possibility needs to be further explored.

The same phenomenon was studied in sperm bound to the egg plasma membrane to evaluate if the rise in $[Ca^{2+}]_i$ also occurs at the time of fusion. Remarkably, we noticed that most of the bound sperm that ended up fusing with the eggs displayed high levels of $[Ca^{2+}]_i$ in both the head and the

flagellum. The midpiece contraction and the immobilization occurred when the levels of $[Ca^{2+}]_i$ went down in the midpiece, suggesting a possible connection between both events. However, our experimental approach limits the interpretation of this result. One possible explanation is that as soon as sperm bind to the plasma membrane of the oocyte, there is a rapid increase in $[Ca^{2+}]_i$ in the sperm. On the other hand, a massive transport of $Ca^{2+}$ from the egg to the sperm could also occur. These hypotheses, however, are hindered by the technical limitations mentioned above. As the oocyte is not loaded with Fluo4, we cannot rule out that the apparent $[Ca^{2+}]_i$ decrease seen in these experiments is due to dye diffusion into the oocyte. Either way, it is clear that the transport of this ion into or out of the sperm is key to cease motility at this fundamental step of fertilization (*Sánchez-Cárdenas et al., 2014*). Additionally, these results demonstrate that only sperm with elevated $[Ca^{2+}]_i$ are capable of binding to the eggs. All these possible scenarios need to be determined in future experiments. Thus, regardless of the mechanism, a clear change in $[Ca^{2+}]_i$ dynamics is observed at the time of midpiece contraction. Future studies will aid to indicate if certain $Ca^{2+}$-dependent proteins that can modify the actin cytoskeleton are responsible for this change.

Why sperm stop moving? we propose three possible hypotheses. The cessation of sperm motility can be attributed to the simultaneous or not occurrence of various events. (1) A rapid increase in $[Ca^{2+}]_i$ levels may trigger the activation of $Ca^{2+}$ pumps within the flagellum. This process consumes local ATP levels, disrupting glycolysis in the process. (2) Reorganization of the actin cytoskeleton: alterations in the actin cytoskeleton can lead to changes in the mechanical properties of the flagellum, impacting its ability to move effectively. (3) Midpiece contraction: contraction in the midpiece region can potentially interfere with mitochondrial function, thereby impeding the energy production necessary for sustained motility. In addition, we speculate that the folding of the flagellum during fusion further facilitates sperm immobilization because it makes it more difficult for the flagellum to beat. Such process can enhance stability and increase the probability of fusion success. Mechanistically, the folding may occur as a consequence of the deformation-induced stress that develops during the decrease of midpiece diameter.

In cilia, apart from the findings presented in this article, nothing is known about the regulation of motility by actin. Actin has been found to participate in ciliogenesis, but its involvement in active motility regulation has not been reported. This highlights a potentially unique role for the actin cytoskeleton in regulating sperm function during fertilization. Our discovery of actin's dynamic reorganization in sperm suggests it could have a more active role in regulating motility and other functions. Given the conservation of cilia and flagella across various organisms, our discovery could prompt further research on the role of the actin network in these structures.

In conclusion, we demonstrate that sperm undergo a structural reorganization of the actin cytoskeleton during key events of fertilization, as summarized in the working model shown in *Figure 9*. Our findings introduce a new aspect of study in reproductive biology. Previous research has mainly focused on identifying proteins essential for sperm-egg fusion. Our results reveal a previously unexplored biological mechanism in mammalian fertilization, opening new avenues for contraceptive method development.

## Materials and methods

For sample details, optical equipment, imaging conditions and probes used, see *Supplementary file 1*.

### Reagents and chemical sources

The chemicals used in this study were procured from the following sources: progesterone, laminin, and concanavalin A were obtained from Sigma-Aldrich Chemical Co. (St. Louis, MO). Fluo-4 AM, pluronic acid, Alexa Fluor-647-phalloidin, FM4-64, FM1-43, Hoechst 33342, and Bodipy-GM1 were acquired from Invitrogen, Thermo Fisher Scientific (Waltham, MA). SiR-Actin and Memglow 700 were sourced from Cytoskeleton (Denver, CO), while ionomycin was purchased from Cayman Chemicals (Ann Arbor, MI). All other chemicals utilized in this research were of reagent grade.

### Animals and housing conditions

Mature mice (8–12-week-old) from the following strains were used in this study: CD1, hybrid F1 (Balb/C x C57BL/6), and transgenic B6D2F1-Tg (CAG/mt-DsRed2, Acr-EGFP) RBGS002Osb (*Hasuwa et al.,*

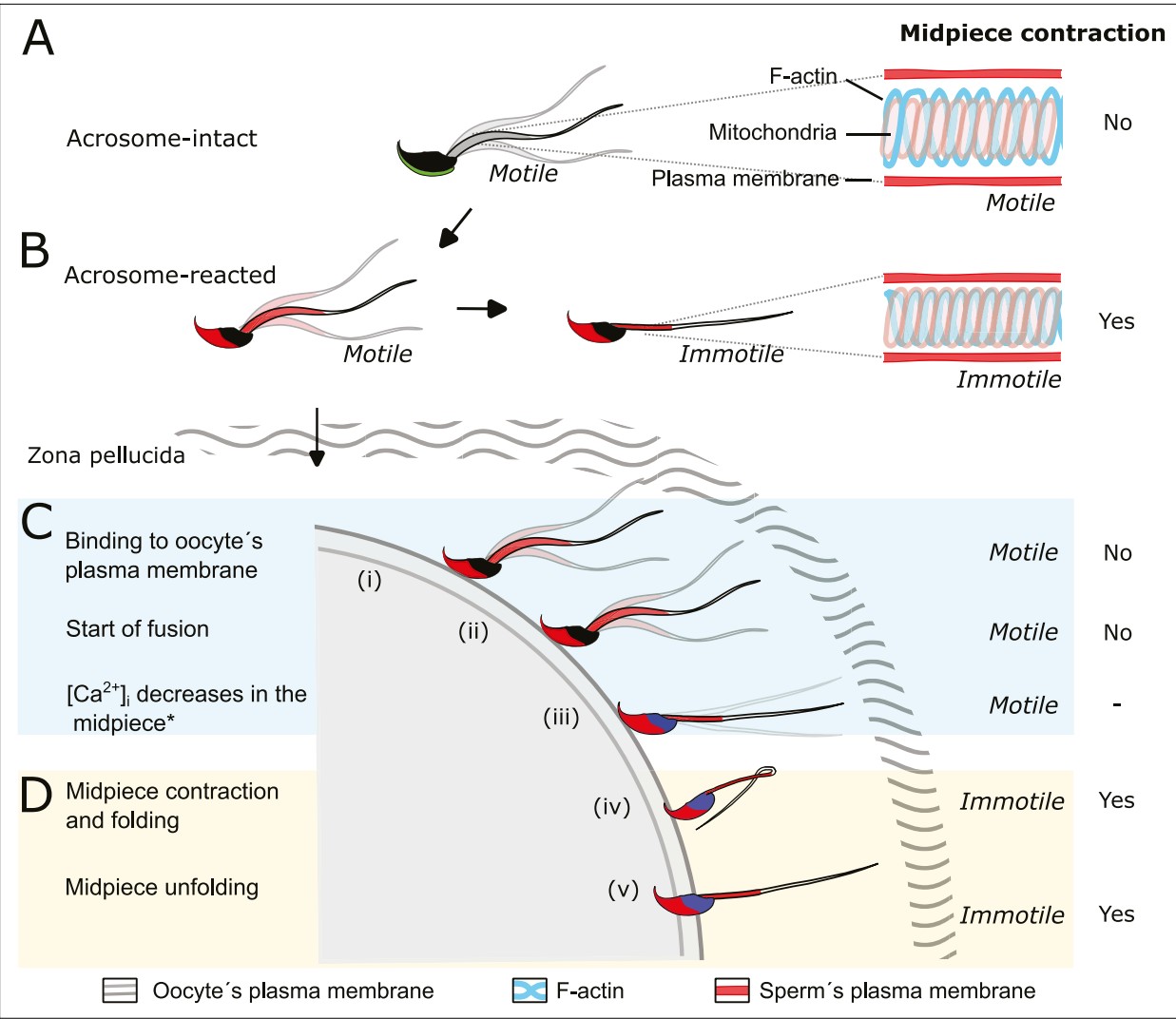

**Figure 9.** Proposed model of the structural reorganization of the sperm actin cytoskeleton during key events of fertilization. The double helix actin network surrounding the mitochondrial sheath of the midpiece undergoes structural changes prior to the motility cessation. This structural modification is accompanied by a decrease in diameter of the midpiece and is driven by intracellular calcium changes that occur concomitant with a reorganization of the actin helicoidal cortex. Although midpiece contraction may occur in a subset of cells that undergo acrosomal exocytosis (AE) (**A, B**), the midpiece contraction occurs prior to motility cessation observed after sperm-egg fusion (**C, D**).

*2010*). Mice were housed in groups of four in a temperature-controlled environment maintained at 23°C, with a light cycle from 07:00 to 19:00 h. The animals had ad libitum access to tap water and laboratory chow.

All experimental procedures adhered to the guidelines of the Institutional Animal Care and were reviewed and approved by the Ethical Committee of the Instituto de Biotecnología, UNAM, and the Instituto de Biología y Medicina Experimental, Buenos Aires (#05/2023). The experiments were conducted in strict accordance with the Guide for Care and Use of Laboratory Animals approved by the National Institutes of Health (NIH).

## Sperm medium and capacitation

The non-capacitating medium employed in this study was a modified TYH medium, which consisted of 119.3 mM NaCl, 4.7 mM KCl, 1.71 mM $CaCl_2 \cdot 2H_2O$, 1.2 mM $KH_2PO_4$, 1.2 mM $MgSO_4 \cdot 7H_2O$, 0.51 mM sodium pyruvate, 5.56 mM glucose, 20 mM 4-(2-hydroxyethyl) piperazine-1-ethanesulfonic acid (HEPES), and 10 μg/ml gentamicin (NC medium). To create capacitating conditions, 15 mM $NaHCO_3$

and 5 mg/ml BSA were added to the medium (CAP medium). In all experiments, the pH was adjusted to 7.4 using NaOH.

The animals were euthanized, and cauda epididymal sperm were harvested. Both cauda epididymides were placed in 1 ml of NC medium. After a 15 min incubation at 37°C (swim-out), the epididymides were removed, and the sperm were resuspended to a maximum final concentration of $10^7$ cells/ml in 100 µl of the appropriate medium. Subsequently, an equal volume (100 µl) of either NC medium or twofold-concentrated capacitating medium (30 mM NaHCO$_3$ and 10 mg/ml BSA) was added, and the sperm were incubated for 60 min at 37°C.

### Live imaging of transgenic mice sperm undergoing AE

Once capacitated, sperm from transgenic B6D2F1-Tg mice were immobilized on coverslips coated with either concanavalin A or laminin (1 mg/ml) to facilitate recordings. Unattached spermatozoa were gently washed away, and the chamber was filled with the recording medium (NC medium) containing 10 µM FM4-64. The application of this dye for continuous AE recording and simultaneous measurement of other parameters, such as $[Ca^{2+}]_i$ concentration, has been previously established (*Sánchez-Cárdenas et al., 2014*). Progesterone (with a stock concentration of 50 mM dissolved in dimethyl sulfoxide [DMSO]), ionomycin (with a stock concentration of 2 mM dissolved in DMSO), or vehicle (DMSO) was prepared in the recording medium and gently applied using a micropipette.

### Sperm motility analysis

For imaging, capacitated sperm from transgenic B6D2F1-Tg mice were immobilized on laminin-coated coverslips (1 mg/ml). Recordings were performed as described earlier using a recording medium (NC medium) containing 10 µM FM4-64. The beating frequency of sperm flagella was calculated using the method proposed by *Corkidi et al., 2021*. The flagella of individual sperm cells were tracked from recorded videos (2 min duration at 100 frames/s) for each experimental condition. The mean orientation of the flagella was detected to infer their beating frequency through the Fourier transform. Each condition was assessed before and after the addition of progesterone, over a total duration of 20 min (with 2 min videos captured every 5 min).

### Scanning electron microscopy

For the fixation of mouse spermatozoa, we utilized the following reagents and media: 0.1 M sodium cacodylate buffer, pH 7.4; TAGA mix, which contains 3.5% glutaraldehyde and 0.5% tannic acid. Coverslips were cleaned by immersing them in 70% ethanol, followed by a mixture of 1 M hydrochloric acid and 50% ethanol for 1 hr, and rinsed with Milli-Q water. Subsequently, they were dried and subjected to plasma cleaning in a Plasma Etch Inc PE-25 Ver 1002 oven under specific conditions: vacuum set point of 150 mTorr, atmospheric vent for 45 s, purge vent for 5 s, gas stabilization for 15 s, and a vacuum alarm set to 3 min. Spermatozoa were obtained by dissecting the cauda epididymis from mice and resuspending them in 1 ml of NC TYH medium. The samples were incubated at 37°C for 15 min, after which 400 µl of medium was collected into two separate tubes. One tube was adjusted to a final volume of 800 µl with 2× CAP medium to achieve 1× CAP medium and incubated for 60 min, followed by the induction of the acrosome reaction using 30 µl of ionomycin for 30 min. Both tubes were centrifuged at 300 × *g* for 3 min, and the supernatant was carefully removed, leaving 100 µl. For fixation, the samples were diluted with 100 µl of PBS and 600 µl of 0.1 M sodium cacodylate buffer and placed on coverslips. After a 10 min incubation, the coverslips were washed three times with 0.1 M sodium cacodylate buffer, each involving shaking in a digital rotator at 80 RPM for 10 min. Subsequently, 100 µl of TAGA mix was added to each coverslip for 15 min. Finally, the coverslips were preserved by submerging them in 0.1 M sodium cacodylate buffer within a Petri dish until further analysis using an Auriga-FIB-Zeiss scanning electron microscope.

To facilitate the quantification of the midpiece diameter of the mouse sperm flagellum, we manually annotated 157 well-focused images. These annotations were used to create a comprehensive image dataset specifically for measuring the midpiece. We employed the ImageJ software for the annotation process. Using the 'Polygon Selection Tool', we carefully delineated the midpiece of each flagellum. Each image was consistently aligned so that the sperm head was positioned to the left, ensuring the midpiece was as straight as possible for accurate measurement. A polygon was drawn around the edges of the midpiece to define a region of interest (ROI). These ROIs were then saved in a

compressed ZIP format. Subsequently, these saved ROIs were imported into a Python script executed in a Jupyter Notebook. This script enabled us to compute the average diameter of the midpiece across all images in the dataset. By leveraging Python's image processing libraries, we automated the calculation, ensuring precision and reproducibility in our measurements.

### Imaging flow cytometry

For the analysis of mouse spermatozoa under NC conditions using image cytometry, the viability dye, Sytox Blue, was procured from Invitrogen, Thermo Fisher Scientific. Transgenic B6D2F1-Tg (CAG/mt-DsRed2, Acr-EGFP) RBGS002Osb mice were used. Sperm were retrieved by sacrificing mice and dissecting the cauda epididymides, which were incubated in 1 ml of NC TYH medium at 37°C for 15 min (swim-out). After incubation, approximately 800 µl of medium containing the sperm was collected, centrifuged at 3000 RPM for 3 min, and the volume was reduced to 300 µl to concentrate the cells. The samples were kept on ice to preserve viability. For fluorescence labeling, Sytox Blue stock solution (1 mM in DMSO) was diluted to 50 µM in TYH medium, and further adjusted to 1 µM with sperm samples using a final volume of 60 µl, mixing 58.8 µl of sperm suspension with 1.2 µl of Sytox Blue solution. Image acquisition was performed using an AMNIS imaging system with a ×60 objective (NA 0.9) and lasers set at 120 mW (488 nm) and 140 mW (561 nm). Sperm images were captured for 30 min, focusing on cells using Aspect Ratio vs. Area in brightfield and RMS Gradient for in-focus selection. Segmentation and analysis of populations of interest were conducted in IDEAS software, where masks for brightfield and fluorescence images were applied to isolate live and dead populations based on Sytox Blue and acrosome fluorescence intensity (EGFP). The resulting data were exported as 16-bit raw images for further analysis.

### Single-cell live imaging of midpiece membrane during AE

Capacitated CD1 mouse sperm were recorded using the previously described method. The recording medium (NC medium) containing 0.5 µM FM4-64 was utilized for this process. The midpiece diameter and fluorescence intensity were quantified using ImageJ software version 1.47V (National Institute of Health, USA). For fluorescence intensity measurement, ROIs were designated in the sperm head and midpiece. After subtracting the background, the formula $(F-F_0)/F_0$ was applied, where $F_0$ represents the baseline calculated by averaging the frames prior to stimulus application.

### Analysis of diameter and FM4-64 fluorescence intensity using kymographs

Kymographs were constructed with the ImageJ Kymograph plug-in. For super-resolution kymographs, a cross line was plotted every 2.5 µm along the midpiece in SRRF images. The R language environment (R 3.3.3 GUI 1.69 Mavericks build, 7328) was employed to obtain diameter values. The super-resolution kymographs were processed using the autocorrelation (acf) function to determine the positions of fluorescence maxima on both sides of the midpiece membrane. The distance between the maxima represented the diameter value. These values were then normalized to those obtained prior to the addition of AE stimulants or vehicle. For fluorescence kymographs, a line was drawn along the midpiece in Total Internal Reflection (TIRF) images. The fluorescence values were also acquired using the R language environment. After subtracting the background, the formula $(F-F_0)/F_0$ was applied, where $F_0$ denotes the baseline, calculated by averaging the frames before stimulus application.

### Correlation between FM4-64 midpiece fluorescence and normalized diameter: Data transformation from fluorescence to contraction

The kymograph analysis provided two main values for each time point evaluated: normalized diameter values every 2.5 µm from the super-resolution kymographs, and normalized fluorescence values for each pixel analyzed (one value every 0.117 µm) from the fluorescence kymographs. To examine the relationship between these values, a graph was created in the R language environment that included all data points, encompassing all analyzed cells, all diameter measurements (every 2.5 µm), and their corresponding fluorescence values for all time points evaluated. In this graph, the x-axis represents the normalized diameter, while the y-axis represents the normalized fluorescence (*Figure 2—figure supplement 1D*). Additionally, a linear regression was performed using the general formula *mx + b*, where *m* is the slope with a value of 8.90 ± 0.15, *b* is the y-intercept with a value of –8.05 ± 0.17,

and R2 is 0.32. With the parameters of the linear regression, we could approximate the midpiece diameter value based on a given FM4-64 fluorescence value. Despite data dispersion, this method provided an approximate value that could indicate whether the midpiece was contracting or not. It is noteworthy that the observed data dispersion may have biological significance at the cell population level, where heterogeneity among individual spermatozoa may contribute to variability in their physiological responses.

## Live imaging of $[Ca^{2+}]_i$ levels and AE

In this study, both CD1 and hybrid F1 mouse sperm were utilized. After undergoing capacitation, the sperm were centrifuged at $300 \times g$ for 4 min. Subsequently, the cells were incubated in NC medium for 20 min with the addition of 1 µM Fluo-4 AM and 0.05% pluronic acid. After the incubation, the cells were centrifuged at $300 \times g$ for 4 min and then resuspended in 200 µl of CAP medium. Recordings were conducted using a previously described method, with the recording medium (NC medium) containing FM4-64 concentrations ranging from 0.5 to 10 µM. For kymograph-like analysis, ROIs were established throughout the midpiece during experiments performed at 10× magnification. $[Ca^{2+}]_i$ levels are represented as $(F-F_0)/F_0$ ratios after background subtraction, where $F_0$ refers to the baseline determined by averaging the frames prior to stimulus application.

## Construction of a 3D kymograph

The 3D kymograph was generated by merging the Fluo4 fluorescence kymograph (constructed in the same way as previously detailed for FM4-64) and the normalized diameter kymograph. Both kymographs possess three dimensions: time (x-axis), length along the midpiece (y-axis), and a color code to distinguish fluorescence intensity/diameter. By utilizing the plot_ly function of the R language environment, the 3D kymograph was plotted with time as the x-axis, length along the midpiece as the y-axis, and fluorescence intensity/diameter as the z-axis. While the color code for this dimension is not required, it remains present to facilitate visual interpretation of the data.

## Single-cell live imaging of midpiece membrane and F-actin during AE

Following swim-out, CD1 mouse sperm were incubated with 100 nM SiR-actin for 10 min in NC medium. To maintain a constant signal and prevent the probe from interfering with actin dynamics, the concentration of SiR-actin remained at 100 nM throughout the entire experiment. Recordings were performed as described earlier, using a recording medium (NC medium) containing 0.5 µM FM4-64 and 100 nM SiR-actin.

## Fluorescence fluctuations super-resolution microscopy

In this study, we employed fluorescence fluctuation super-resolution microscopy to process FM4-64 and SiR-actin images. The SRRF approach was utilized, leveraging the NanoJ plug-in within FIJI/ImageJ software (*Gustafsson et al., 2016*; *Culley et al., 2018*). Each image consisted of 100 temporal frames, acquired with an exposure time of 10 ms per frame. During the analysis, specific parameters were set as follows: ring radius at 0.5, radiality magnification at 5, axes in ring at 8, and batch processing enabled. All other parameters were maintained at their default values.

Additionally, Memglow 700, Bodipy-GM1, and FM1-43 images were processed using the MSSR algorithm, facilitated by the MSSR plug-in within FIJI/ImageJ (*Torres-García et al., 2022*). Similar to the SRRF approach, each image was composed of 100 temporal frames, with an exposure time of 10 ms per frame. For the analysis, the following parameters were applied: AMP set to 5, PSF set to 10, order set to 0, and the interpolation type designated as bicubic.

## Colocalization analysis using Manders' coefficients

To quantitatively evaluate the spatial overlap and potential associations between the fluorescently labeled biological structures, a colocalization analysis was conducted using Manders' colocalization coefficients (*Dunn et al., 2011*). This analysis focused on the colocalization of FM4-64 and SiR-actin signals within the region of the midpiece of the flagellum, where the contraction initiation was observed.

Manders' colocalization coefficients (M1 and M2) were calculated using the R programming language. These coefficients provide a quantitative measure of colocalization between two

fluorophores, with values ranging from 0 (indicating no colocalization) to 1 (representing complete colocalization). M1 corresponds to the fraction of intensity in channel 1 that overlaps with the intensity in channel 2, while M2 denotes the fraction of intensity in channel 2 that overlaps with the intensity in channel 1.

The utilization of Manders' coefficients offers advantages such as decreased sensitivity to variations in signal intensities and increased robustness against changes in background noise, as compared to other colocalization measurements, including Pearson's correlation coefficient and overlap coefficients. By determining the Manders' colocalization coefficients for the super-resolution images of the region of the flagellum's midpiece where the contraction initiation was observed, we objectively assessed the extent of spatial overlap between FM4-64 and SiR-actin. This analysis yielded valuable insights into potential interactions or associations between these two structures, ultimately contributing to an enhanced understanding of the underlying biological processes.

## Analysis of fluorescence peak position changes to assess relative proximity between plasma membrane and actin cytoskeleton

The aim of this study was to assess the relative proximity between the plasma membrane (labeled with FM4-64) and the actin cytoskeleton (labeled with SiR-actin) by analyzing the position changes of fluorescence peaks over time. SRRF images were first generated for both structures. Subsequently, kymographs were computed from these SRRF images, and the spatial derivatives of the kymographs were calculated to identify the fluorescence peaks.

The R programming language was utilized for data analysis, computing the spatial derivatives of the kymographs to track the positions of the peaks. The fluorescence peak was identified at the location where the derivative transitioned between positive and negative values. The center of the cell was determined by averaging the positions of both membrane peaks, and subsequently, it was normalized to 0. Given the symmetrical nature of the fluorescence peaks around the cell center, the analysis focused on the right side of the graph (right peaks for FM4-64 and SiR-actin).

The distances to the peaks were plotted over time, followed by a linear regression. By examining the relative positions between the fluorescence maxima and their changes over time, the goal was to gain insights into the spatial relationship and potential interactions between the plasma membrane and the actin cytoskeleton.

## Three-dimensional stochastic optical reconstruction microscopy (3D-STORM)

Following incubation under appropriate conditions (NC with DMSO or CAP with 100 µM progesterone), cells were washed with NC medium via centrifugation (4 min at 400 × $g$) and resuspended in NC medium. Sperm were seeded onto poly-L-lysine-coated coverslips (Corning #1.5), air-dried for 10 min, fixed, and permeabilized with 0.3% fresh glutaraldehyde and 0.25% Triton-X100 in cytoskeleton buffer (CB, containing 10 mM MES, 150 mM NaCl, 5 mM EGTA, 5 mM glucose, and 5 mM MgCl$_2$, pH = 6.1) for 1 min at room temperature. This was followed by three washes with CB (5 min at room temperature). Next, cells were incubated with 2% glutaraldehyde in CB for 15 min at room temperature and washed twice with CB (10 min at room temperature). Samples were treated with 0.1% NaBH$_4$ (freshly prepared in PBS) for 7 min at room temperature to reduce background fluorescence, washed twice with PBS (5 min at room temperature), and incubated for 1 hr at room temperature with Alexa 568-Peanut agglutinin (PNA, 0.01 mg/ml) and Alexa 647-Phalloidin (1:13) diluted in PBS. Sperm were then washed with PBS for 5 min and immediately mounted in STORM imaging buffer (50 mM Tris-HCl pH 8, 10 mM NaCl, 0.56 mg/ml glucose oxidase, 34 µg/ml catalase, 10% glucose, and 1% β-mercaptoethanol). Nonspecific staining was determined by incubating sperm in the absence of phalloidin (*Gervasi et al., 2018*).

Images were acquired using Andor IQ 2.3 software on a custom-built microscope equipped with an Olympus PlanApo 100× NA 1.45 objective and a CRISP ASI autofocus system (*Campagnola et al., 2015*; *Weigel et al., 2011*). Alexa Fluor 647 was excited with a 642 nm laser (DL640-150-O, CrystaLaser, Reno, NV) under continuous illumination. Initially, the photo-switching rate was sufficient to provide a substantial fluorophore density. However, as fluorophores irreversibly photo-bleached, a 405 nm laser was introduced to enhance photo-switching. The intensity of the 405 nm laser was adjusted within the range of 0.01–0.5 mW to maintain an appropriate density of active fluorophores.

Axial localization was achieved through astigmatism using a MicAO 3DSR adaptive optics system (Imagine Optic, Orsay, France), which allowed for both the correction of spherical aberrations and the introduction of astigmatism (*Izeddin et al., 2012*). Axial localization with adaptive optics enabled 3D reconstruction over a thickness of 1 µm (*Gervasi et al., 2018*). A calibration curve for axial localization was generated with 100 nm TetraSpeck microspheres (Invitrogen) immobilized on a coverslip (*Huang et al., 2008*). Images were acquired using a water-cooled, back-illuminated EMCCD camera (Andor iXon DU-888) operated at –85°C at a rate of 23 frames/s. A total of 50,000 frames were collected to generate each super-resolution image. Single-molecule localization, drift correction using image cross-correlation, and reconstruction were performed with ThunderSTORM (*Ovesný et al., 2014*). To analyze the molecular radial distributions, ROIs of the flagellum were selected, ensuring they were in a straight line. The center of the flagella cross-section was initially identified by Gaussian fitting of the localization histograms along the x and y axes. Subsequently, the coordinates of the localized molecules were transformed into cylindrical coordinates to obtain the radial position ($r$) and azimuthal angle ($\Theta$) (*Gervasi et al., 2018*; *Luque et al., 2021*; *Stival et al., 2018*). This approach allowed for a detailed examination of the spatial organization and molecular distribution of the sperm cell structures.

## Number and brightness

To investigate structural rearrangements of the actin cytoskeleton during AE, a moment-based analysis was performed (*Digman et al., 2008*). This approach involved processing fluorescence signal intensity fluctuations to gain insight into actin filament polymerization and redistribution in the midpiece. Time-lapse images of live sperm stained with SiR-actin during AE induction were captured, with 100 frames acquired every 30 s. For each consecutive 100-frame segment of the time-lapse, temporal signal mean and variance were calculated, and number and brightness maps, as well as mean intensity sequences, were obtained (*Figure 5—figure supplement 3A*).

Prior to this step, a sCMOS sensor noise-correction algorithm (*Mandracchia et al., 2020*) was applied to the images to reduce the effect of local digital noise variation on the analysis. Camera calibration maps were obtained as described elsewhere (*Liu et al., 2017*). Extended spatial resolution micrographs of the time lapses were obtained using MSSR, which allowed for the identification of nanoscale local enrichments of polymerized actin in the midpiece. Locations of these enrichments were selected and stored using the ROI tool in FIJI, with obtained ROIs consisting of 1 µm² areas over the identified local SiR-actin signal enrichment zones. By using the 'RImajeJROI' package for RStudio, saved ROIs were imported into the R environment, where they were utilized to extract information on the dynamic changes of actin signal as a consequence of AE induction (*Figure 5—figure supplement 3B*). This analysis provided valuable insights into the structural organization and dynamics of the actin cytoskeleton during the AE process.

## IVF assay

Here, 8–12-week-old F1 female mice were superovulated using equine chorionic gonadotropin (5 IU, PMSG; Syntex, Argentina) administered at 18:30, followed by human chorionic gonadotropin (5 IU, hCG; Syntex) intraperitoneal injection 48 hr later. Cumulus oocyte complexes were collected from oviducts 12–13 hr post-hCG administration and placed in TYH IVF medium (which contains 25 mM NaHCO$_3$ and 4 mg/ml BSA, without HEPES addition). The complexes were then inseminated with capacitated sperm at a final concentration ranging between $1 \times 10^6$ and $2 \times 10^6$ sperm/ml (a high concentration to allow polyspermy).

Following a 4 hr coincubation period at 37°C with 5% CO$_2$, oocytes were stained with Hoechst 33342 (10 µg/ml) for 15 min and subsequently washed three times before being mounted on a glass slide. Oocytes were placed on 5 µl of CAP 0.5× modified TYH medium (7.5 mM NaHCO$_3$ and 2.5 mg/ml BSA, 10 mM HEPES) containing 10 µM FM4-64, gently compressed beneath an $18 \times 18$ mm² coverslip supported by four solid Vaseline spots, and sealed with nail polish. This IVF assay protocol facilitated the study of fertilization events and sperm-oocyte interactions in a controlled in vitro environment.

## Sperm-oocyte fusion assay

The sperm-oocyte fusion event was assessed using the zona-free oocyte penetration assay in conjunction with the Hoechst dye transfer technique (*Conover and Gwatkin, 1988*; *Hinkley et al., 1986*; *Stewart-Savage and Bavister, 1988*). Cumulus oocyte complexes were collected as previously

described. Oocytes were freed from cumulus cells by treatment with 0.2 mg/ml hyaluronidase and had their ZP removed by exposure to acid Tyrode's solution (pH 2.5) for 10–20 s (*Nicolson et al., 1975*).

Zona-free mouse oocytes were preloaded with 1 µg/ml Hoechst 33342 in TYH IVF medium for 5 min at 37°C. Following incubation, oocytes were washed three times for 20 min each in fresh TYH IVF medium and mounted in an oocyte-holding Petri dish (*Jin et al., 2011*; MatTek glass bottom dishes, 35 mm Petri dish, 14 mm Microwell poly-D-lysine-coated P35GC-1.5-14C). Zona-free oocytes were placed on 5 µl of TYH IVF medium containing 10 µM FM4-64 and gently compressed under a 9 × 9 mm² coverslip supported by four solid Vaseline spots. TYH IVF medium containing 10 µM FM4-64 was added until reaching a final volume of 200 µl, and the medium was then covered with prewarmed (37°C) mineral oil (M8410, Sigma).

The dish was placed into the microscope incubation chamber at 37°C with 5% $CO_2$, and oocytes were inseminated with capacitated sperm at a final concentration of $1 \times 10^5$ cells/ml. For *Figure 7*, EGFP-DsRed2 sperm were used, while for *Figure 8*, F1 sperm loaded with Fluo4 (as described above) were utilized. This assay allowed for the investigation of sperm-oocyte fusion events and the assessment of the transfer of Hoechst dye between the two cell types.

## Statistical analysis

Data are presented as mean ± standard error of the mean (SEM) from a minimum of three independent experiments for all measurements. Statistical analyses were conducted using GraphPad Prism v4.0 (San Diego, CA) or the R language environment (R 3.3.3 GUI 1.69 Mavericks build [7328]). The specific statistical analysis employed is indicated in the relevant figure legends. A p-value of $<0.05$ was considered statistically significant.

## Acknowledgements

We gratefully acknowledge the financial support provided by the Williams and Rene Baron Foundations, the Male Contraceptive Initiative (MCI), and the Chan-Zuckerberg Initiative (CZI) for this work. Our sincere thanks go to Dr. Pablo Visconti for his valuable insights throughout the course of this project. We also thank Yoloxochitl Sánchez-Guevara for her technical support. This paper is dedicated to the memory of OAP. This work was supported by Chan Zuckerberg Initiative (2021-240504 to MB and AG, GBI-0000000093 to AG, 2022-252509 to AG); Agencia Nacional de Promoción Científica y Tecnológica (PICT, 2017-3047, 2018-1988, and 2020-00988); Dirección General de Asuntos del Personal Académico/Universidad Nacional Autónoma de México (DGAPA/UNAM grant: IN105222 to GK, IN211821 to AG and IN200919 to AD); and National Institute of Health (R01HD380882 to AD, R01HD106968 to DK and MGB).

## Additional information

### Funding

| Funder | Grant reference number | Author |
| --- | --- | --- |
| Chan Zuckerberg Initiative | 2021-240504 | Adán Guerrero<br>Mariano G Buffone |
| Chan Zuckerberg Initiative | GBI-0000000093 | Adán Guerrero |
| Agencia Nacional de Promoción de la Investigación, el Desarrollo Tecnológico y la Innovación | 2020-00988 | Mariano G Buffone |
| Eunice Kennedy Shriver National Institute of Child Health and Human Development | R01HD106968 | Diego Krapf<br>Mariano G Buffone |

| Funder | Grant reference number | Author |
| --- | --- | --- |
| Agencia Nacional de Promoción de la Investigación, el Desarrollo Tecnológico y la Innovación | 2017-3047 | Dario Krapf |
| Chan Zuckerberg Initiative | 2022-252509 | Adán Guerrero |
| Agencia Nacional de Promoción Científica y Tecnológica | PICT | Mariano G Buffone |
| Agencia Nacional de Promoción Científica y Tecnológica | 2018-1988 | Mariano G Buffone |
| Dirección General de Asuntos del Personal Académico, Universidad Nacional Autónoma de México | IN105222 | Gabriel Corkidi |
| Dirección General de Asuntos del Personal Académico, Universidad Nacional Autónoma de México | IN211821 | Adán Guerrero |
| Dirección General de Asuntos del Personal Académico, Universidad Nacional Autónoma de México | IN200919 | Alberto Darszon |
| National Institutes of Health | R01HD380882 | Alberto Darszon |

The funders had no role in study design, data collection and interpretation, or the decision to submit the work for publication.

## Author contributions

Martina Jabloñski, Data curation, Investigation, Writing – original draft; Guillermina M Luque, Matias Gomez Elias, Claudia Sanchez Cardenas, Jose L de La Vega Beltran, Alejandro Linares, Investigation; Xinran Xu, Gabriel Corkidi, Aquetzalli Arenas-Hernandez, María DP Ramos-Godinez, Methodology; Victor Abonza, Data curation; Alejandro López-Saavedra, Resources; Dario Krapf, Alberto Darszon, Conceptualization; Diego Krapf, Conceptualization, Methodology; Adán Guerrero, Conceptualization, Supervision, Writing – original draft; Mariano G Buffone, Conceptualization, Supervision, Writing – original draft, Writing - review and editing

## Author ORCIDs

Martina Jabloñski  http://orcid.org/0000-0001-6465-0609
Guillermina M Luque  http://orcid.org/0000-0002-2745-8236
Dario Krapf  https://orcid.org/0000-0001-7607-1954
Mariano G Buffone  https://orcid.org/0000-0002-7163-6482

## Ethics

All experimental procedures adhered to the guidelines of the Institutional Animal Care and were reviewed and approved by the Ethical Committee of the Instituto de Biotecnología, UNAM, and the Instituto de Biotecnología y Medicina Experimental, Buenos Aires. The experiments were conducted in strict accordance with the Guide for Care and Use of Laboratory Animals approved by the National Institutes of Health (NIH).

Reviewer #2 (Public Review): https://doi.org/10.7554/eLife.93792.3.sa1
Reviewer #3 (Public Review): https://doi.org/10.7554/eLife.93792.3.sa2
Author response https://doi.org/10.7554/eLife.93792.3.sa3

# Additional files

## Supplementary files
- Supplementary file 1. Sample details, optical equipment, imaging conditions, and probes used.
- MDAR checklist

## Data availability
Data acquired and analyzed to prepare the figures during this study are included in this manuscript is available at Zenodo and the code can be found at GitHub (copy archived at *Jabloński, 2024*).

The following dataset was generated:

| Author(s) | Year | Dataset title | Dataset URL | Database and Identifier |
|---|---|---|---|---|
| Jabloński M, Luque G, Gómez-Elías M, Sanchez-Cardenas C, Xu X, de la Vega-Beltran JL, Corkidi G, Linares A, Abonza Amaro VX, Arenas-Hernandez A, Ramos-Godinez MDP, López-Saavedra A, Krapf D, Krapf D, Darszon A, Guerrero A, Buffone MG | 2024 | Reorganization of the Flagellum Scaffolding Induces a Sperm Standstill During Fertilization | https://doi.org/10.5281/zenodo.13769608 | Zenodo, 10.5281/zenodo.13769608 |

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
