## [Editor Report · eLife assessment]

This **important** work substantially advances our understanding of sperm motility regulation during fertilization process by uncovering the midpiece/mitochondria contraction associated with motility cessation and structural changes in the midpiece actin network as its mode of action involved. The evidence supporting the conclusion is **solid**, with rigorous live-cell imaging using state-of-the-art microscopy, although more functional analysis of the midpiece/mitochondria contraction would have further strengthened the study. The work will be of broad interest to cell biologists working on the cytoskeleton, mitochondria, cell fusion, and fertilization.

---

## [Referee Report · Reviewer #2 (Public Review)]

Summary:

The authors used state-of-the-art microscopy to analyze the structural changes that occur in sperm tails after the acrosome reaction. They found that midpiece contraction and actin reorganization occurred, which is associated with the cessation of flagellar motility during sperm-egg fusion. The mechanism by which flagellar motility is arrested during sperm-oocyte fusion is unknown, and this study proposes its novel mechanism and provides important insights for cell and reproductive biologists.

In the revised manuscript, the authors addressed most of my concerns.

Strength:

Various microscopy techniques including super-resolution microscopy and scanning electron microscopy were used to analyze structural organization of the midpiece in detail.

---

## [Referee Report · Reviewer #3 (Public Review)]

While progressive and also hyperactivated motility are required for sperm to reach the site of fertilization and to penetrate oocyte's outer vestments, during fusion with the oocyte's plasma membrane it has been observed that sperm motility ceases. Identifying the underlying molecular mechanisms would provide novel insights into a crucial but mostly overlooked physiological change during the sperm's life cycle. In this publication the authors aim to provide evidence that the helical actin structure surrounding the sperm mitochondria in the midpiece plays a role in regulating sperm motility, specifically the motility arrest during sperm fusion but also during earlier cessation of motility in a subpopulation of sperm post acrosomal exocytosis.

The main observation the authors make is that in a subpopulation of sperm undergoing acrosomal exocytosis and sperm that fuse with the plasma membrane of the oocyte display a decrease in midpiece parameter of 30 nm. The authors propose the decrease in midpiece diameter via various microscopy techniques based on membrane dyes and bright-field images. In the revised version of the manuscript, a change in midpiece diameter is now confirmed via electron microscopy, even though the difference is not significant. The authors also propose that the midpiece diameter decrease is driven by changes in sperm intracellular Ca2+ and structural changes of the actin helix network. Future studies are still needed to confirm the casualty of these events and explore the discrepancy between fluorescence microscopy results and SEM. Overall, the authors should further tone down their conclusions.

---

## [Author Response]

The following is the authors’ response to the original reviews.

**Reviewer #1 (Public Review):**
Summary:This important work advances our understanding of sperm motility regulation during fertilization by uncovering the midpiece/mitochondria contraction associated with motility cessation and structural changes in the midpiece actin network as its mode of action. The evidence supporting the conclusion is solid, with rigorous live cell imaging using state-of-the-art microscopy, although more functional analysis of the midpiece/mitochondria contraction would have further strengthened the study. The work will be of broad interest to cell biologists working on the cytoskeleton, mitochondria, cell fusion, and fertilization. Strengths: The authors demonstrate that structural changes in the flagellar midpiece F-actin network are concomitant to midpiece/mitochondrial contraction and motility arrest during sperm-egg fusion by rigorous live cell imaging using state-of-art microscopy.

Response P1.1: We thank the reviewer for her/his positive assessment of our manuscript.

Weaknesses:Many interesting observations are listed as correlated or in time series but do not necessarily demonstrate the causality and it remains to be further tested whether the sperm undergoing midpiece contraction are those that fertilize or those that are not selected. Further elaboration of the function of the midpiece contraction associated with motility cessation (a major key discovery of the manuscript) would benefit from a more mechanistic study.

Response P1.2: We thank the reviewer for this point. We have toned down some of our statements since some of the observations are indeed temporal correlations. We will explore some of these possible connections in future experiments. In addition, we have now incorporated additional experiments and possible explanations about the function of the midpiece contraction.

**Reviewer #2 (Public Review):**
(1) The authors used various microscopy techniques, including super-resolution microscopy, to observe the changes that occur in the midpiece of mouse sperm flagella. Previously, it was shown that actin filaments form a double helix in the midpiece. This study reveals that the structure of these actin filaments changes after the acrosome reaction and before sperm-egg fusion, resulting in a thinner midpiece. Furthermore, by combining midpiece structure observation with calcium imaging, the authors show that changes in intracellular calcium concentrations precede structural changes in the midpiece. The cessation of sperm motility by these changes may be important for fusion with the egg. Elucidation of the structural changes in the midpiece could lead to a better understanding of fertilization and the etiology of male infertility. The conclusions of this manuscript are largely supported by the data, but there are several areas for improvement in data analysis and interpretation. Please see the major points below.

Response P2.1: We thank the reviewer for the positive comments.

(2) It is unclear whether an increased FM4-64 signal in the midpiece precedes the arrest of sperm motility. in or This needs to be clarified to argue that structural changes in the midpiece cause sperm motility arrest. The authors should analyze changes in both motility and FM4-64 signal over time for individual sperm.

Response P2.2 : We have conducted single cell experiments tracking both FM4-64 and motility as the reviewer suggested (Supplementary Fig S1). We have observed that in all cases, cells gradually diminished the beating frequency and increased FM4-64 fluorescence in the midpiece until a complete motility arrest is observed. A representative example is shown in this Figure but we will reinforce this concept in the results section.

(3) It is possible that sperm stop moving because they die. Figure 1G shows that the FM464 signal is increased in the midpiece of immotile sperm, but it is necessary to show that the FM4-64 signal is increased in sperm that are not dead and retain plasma membrane integrity by checking sperm viability with propidium iodide or other means.

Response P2.3: This is a very good point. In our experiments, we always considered sperm that were motile to hypothesize about the relevance of this observation. We have two types of experiments:

(1) Sperm-egg Fusion: In experiments where sperm and eggs were imaged to observe their fusion, sperm were initially moving and after fusion, the midpiece contraction (increase in FM4-64 fluorescence was observed) indicating that the change in the midpiece (that was observed consistently in all fusing cells analyzed), is part of the process.

(2) Sperm that underwent acrosomal exocytosis (AE): we have observed two behaviours as shown in Figure 1:

a) Sperm that underwent AE and they remain motile without midpiece contraction (they are alive for sure);

b) Sperm that underwent AE and stopped moving with an increase in FM464 fluorescence. We propose that this contraction during AE is not desired because it will impede sperm from moving forward to the fertilization site when they are in the female reproductive tract. In this case, we acknowledge that the cessation of sperm motility may be attributed to cellular death, potentially correlating with the increased FM4-64 signal observed in the midpiece of immotile sperm that have undergone AE. To address this hypothesis, we conducted image-based flow cytometry experiments, which are well-suited for assessing cellular heterogeneity within large populations.

Author response image 1 illustrates the relationship between cell death and spontaneous AE in noncapacitated mouse sperm, where intact acrosomes are marked by EGFP. Cell death was evaluated using Sytox Blue staining, a dye that is impermeable to live cells and shows affinity for DNA. AE was assessed by the absence of EGFP in the acrosome.

Author response image 1a indicates a lack of correlation between Sytox and EGFP fluorescence. Two populations of sperm with EGFP signals were found (EGFP+ and EGFP-), each showing a broad distribution of Sytox signal, enabling the distinction between cells that retain plasma membrane integrity (live sperm: Sytox-) and those with compromised membranes (dead cells: Sytox+). The observed bimodal distribution of EGFP signal, regardless of live versus dead cell populations, indicates that the fenestration of the plasma membrane known to occur during AE is a regulated process that does not necessarily compromise the overall plasma membrane integrity.

These observations are reinforced by the single-cell examples in Author response image 1b, where we were able to identify sperm in four categories: live sperm with intact acrosome (EGFP+/Sytox-), live sperm with acrosomal exocytosis (EGFP-/Sytox-), dead sperm with intact acrosome (EGFP+/Sytox+), and dead sperm with AE (EGFP-/Sytox+). Note the case of AE (lacking EGFP signal) which bears an intact plasma membrane (lacking Sytox Blue signal). Author response image 2 shows single-cell examples of the four categories observed with confocal microscopy to reinforce the observations from Author response image 1a.

**Author response image 1. sa3fig1:** Fi Image based flow cytometry analysis (ImageStream Merk II), of non-capacitated mouse sperm, showing the distribution of EGFP signal (acrosome integrity) against Sytox Blue staining (cell viability). (A) The quadrants show: Sytox Blue + / EGFP low (17.6%), Sytox Blue + / EGFP high (40.1%), Sytox Blue - / EGFP high (20.2%), and Sytox Blue - / EGFP low (21.7%). Each quadrant indicates the percentage of the total sperm population exhibiting the corresponding staining pattern. Axes are presented in a log10 scale of arbitrary units of fluorescence. (B) Representative single-cell images corresponding to the four categorized sperm populations from the flow cytometry analysis in panel (A). The top row displays sperm with compromised plasma membrane integrity (Sytox Blue +), showing low (left) and high (right) EGFP signals. The bottom row shows sperm with intact plasma membrane (Sytox Blue -), displaying high (left) and low (right) EGFP signal. It is worth noting that when analyzing the percentages in (A), we observed that the data also encompass a population of headless flagella, which was present in all observed categories. Therefore, the percentages should be interpreted with caution.

**Author response image 2. sa3fig2:** Confocal Microscopy Examples of AE and cell viability. The top row features sperm with compromised plasma membrane integrity (Sytox Blue +) and high EGFP expression; the second row displays sperm with compromised membrane and low EGFP expression; the third row illustrates sperm with intact membrane (Sytox Blue -) and high EGFP expression; the bottom row shows sperm with intact membrane and low EGFP expression.

Author response images 3-5 provide insight into the relationship between FM4-64 and Sytox Blue fluorescence intensities in non-capacitated sperm (CTRL, Author response image 3), capacitated sperm and acrosome exocytosis events stimulated with 100 µM progesterone (PG, Author response image 4), and capacitated sperm stimulated with 20 µM ionomycin (IONO, Author response image 5). Two populations of sperm with Sytox Blue signals were clearly distinguished (Sytox+ and Sytox-), enabling the discernment between live and dead sperm. Interestingly, the upper right panels of Author response images 3A, 4A, and 5A (Sytox Blue+ / FM4-64 high) consistently show a positive correlation between FM4-64 and Sytox Blue. This observation aligns with the concern raised by Reviewer 2, suggesting that compromised membranes due to cell death provide more binding sites for FM4-64.

Nonetheless, the lower panels of Author response images 3A, 4A and 5A (Sytox Blue-) show no correlation with FM4-64 fluorescence, indicating that this population can exhibit either low or high FM4-64 fluorescence. As expected, in stark contrast with the CTRL case, the stimulation of AE with PG or IONO in capacitated sperm increased the population of live sperm with high FM4-64 fluorescence (Sytox Blue+ / FM4-64 high: CTRL: 7.85%, PG: 8.73%, IONO: 13.5%).

Single-cell examples are shown in Author response images 3B, 4B, and 5B, where the four categories are represented: dead sperm with low FM4-64 fluorescence (Sytox Blue+ / FM4-64 low), dead sperm with high FM4-64 fluorescence (Sytox Blue+ / FM4-64 high), live sperm with low FM4-64 fluorescence (Sytox Blue- / FM4-64 low), and live sperm with high FM4-64 fluorescence (Sytox Blue- / FM4-64 high).

**Author response image 3. sa3fig3:** Relationship between cell death and FM4-64 fluorescence in capacitated sperm without inductor of RA**.** Image-based flow cytometry analysis of non-capacitated mouse sperm loaded with FM464 and Sytox Blue dyes, with one and two minutes of incubation time, respectively.

**Author response image 4. sa3fig4:** Relationship between cell death and FM4-64 fluorescence capacitated sperm stimulated with progesterone. Image-based flow cytometry analysis of non-capacitated mouse sperm loaded with FM4-64 and Sytox Blue dyes, with one and two minutes of incubation time, respectively. (A) The quadrants show: Sytox Blue+ / FM4-64 low (9.04%), Sytox Blue+ / FM4-64 high (61.6%), Sytox Blue- / FM4-64 low (19.7%), and Sytox Blue- / FM4-64 high (8.73%). Each quadrant indicates the percentage of the total sperm population exhibiting the corresponding staining pattern. Axes are presented on a log10 scale of arbitrary units of fluorescence. (B) Representative single-cell images corresponding to the four categorized sperm populations from the flow cytometry analysis in panel (A)

**Author response image 5. sa3fig5:** Relationship between cell death and FM4-64 fluorescence capacitated sperm stimulated with ionomycin. Image-based flow cytometry analysis of non-capacitated mouse sperm loaded with FM464 and Sytox Blue dyes, with one and two minutes of incubation time, respectively. (A) The quadrants show: Sytox Blue+ / FM4-64 low (4.52%), Sytox Blue+ / FM4-64 high (60.6%), Sytox Blue- / FM4-64 low (20.5%), and Sytox Blue- / FM4-64 high (13.5%). Each quadrant indicates the percentage of the total sperm population exhibiting the corresponding staining pattern. Axes are presented on a log10 scale of arbitrary units of fluorescence. (B) Representative single-cell images corresponding to the four categorized sperm populations from the flow cytometry analysis in panel (A).

Based on the data presented in Author response images 1 to 6, we derive the following conclusions summarized below:

(1) There is no direct relationship between cell death (Sytox Blue-) and AE (EGFP) (Author response images 1 and 2).

(2) There is bistability in the FM4-64 fluorescent intensity. Before reaching a certain threshold, there is no correlation between FM4-64 and Sytox Blue signals, indicating no cell death. However, after crossing this threshold, the FM4-64 signal becomes correlated with Sytox Blue+ cells, indicating cell death (Author response images 4-6).

(3) The Sytox Blue- population of capacitated sperm is sensitive to AE stimulation with progesterone, leading to the expected increase in FM4-64 fluorescence.

Therefore, while the FM4-64 signal alone is not a definitive marker for either AE or cell death, it is crucial to use additional viability assessments, such as Sytox Blue, to accurately differentiate between live and dead sperm in studies of acrosome exocytosis and sperm motility. In the present work, we did not use a cell viability marker due to the complex multicolor, multidimensional fluorescence experiments. However, cell viability was always considered, as any imaged sperm was chosen based on motility, indicated by a beating flagellum. The determination of whether selected sperm die during or after AE remains to be elucidated. The results presented in Figure 2 and Supplementary S1 show examples of motile sperm that experience an increase in FM4-64 fluorescence.

All this information is added to the manuscript (Supplementary Figure 1D).

(4) It is unclear how the structural change in the midpiece causes the entire sperm flagellum, including the principal piece, to stop moving. It will be easier for readers to understand if the authors discuss possible mechanisms.

Response P2.4: As requested, we have incorporated a possible explanation in the discussion section (see line 644-656). We propose three possible hypotheses for the cessation of sperm motility, which can be attributed to the simultaneous occurrence of various events:

(1) Rapid increase in [Ca2+]i levels: A rapid increase in [Ca2+]i levels may trigger the activation of Ca2+ pumps within the flagellum. This process consumes local ATP levels, disrupting glycolysis and thereby depleting the energy required for motility.

(2) Reorganization of the actin cytoskeleton: Alterations in the actin cytoskeleton can lead to changes in the mechanical properties of the flagellum, impacting its ability to move effectively.

(3) Midpiece contraction: Contraction in the midpiece region can potentially interfere with mitochondrial function, impeding the energy production necessary for sustained motility.

(5) The mitochondrial sheath and cell membrane are very close together when observed by transmission electron microscopy. The image in Figure 9A with the large space between the plasma membrane and mitochondria is misleading and should be corrected. The authors state that the distance between the plasma membrane and mitochondria approaches about 100 nm after the acrosome reaction (Line 330 - Line 333), but this is a very long distance and large structural changes may occur in the midpiece. Was there any change in the mitochondria themselves when they were observed with the DsRed2 signal?

Response P2.5: The authors appreciate the reviewer’s observation regarding the need to correct the image in Figure 9A, as the original depiction conveys a misleading representation of the spatial relationship between the mitochondrial sheath and the plasma membrane. This figure has been corrected to accurately reflect a more realistic proximity, while keeping in mind that it is a cartoonish representation.

Regarding the comments about the distances mentioned between former lines 330 and 333, the measurement was not intended to describe the gap between the plasma membrane and the mitochondria but rather the distance between F-actin and the plasma membrane.

Author response image 6 shows high-resolution scanning electron microscopy (SEM) of two sperm fixed with a protocol tailored to preserve plasma membranes (ref), where the insets clearly show the flagellate architecture in the midpiece with an intact plasma membrane covering the mitochondrial network. A non-capacitated sperm with an intact acrosome is shown in panel A, and a capacitated sperm that has experienced AE is shown in panel B.

Notably, the results depicted in Author response image 6 demonstrate that, irrespective of the AE status, the distance between the plasma membrane and mitochondria consistently remains less than 20 nm, thus confirming the close proximity of these structures in both physiological states. As Reviewer 2 pointed out, if there is no significant difference in the distance between the plasma membrane and mitochondria, then the observed structural changes in the actin network within the midpiece should somehow alter the actual deposition of mitochondria within the midpiece. Figure 5D-F shows that midpiece contraction is associated with a decrease in the helical pitch of the actin network; the distance between turns of the actin helix decreases from l = 248 nm to l = 159 nm. This implies a net change in the number of turns the helix makes per 1 µm, from 4 to 6 µm-1.

**Author response image 6. sa3fig6:** SEM image showing the proximity between plasma membrane and mitochondria. Scale bar 100 nm.

Additionally, a structural contraction can be observed in Figure 5D-F, where the radius of the helix decreases by about 50 nm. To clarify this point, we sought to measure the deposition of individual DsRed2 mitochondria using computational superresolution microscopy—FF-SRM (SRRF and MSSR), Structured Illumination Microscopy (SIM), or a combination of both (SIM + MSSR), in 2D. Author response image 7 shows that these three approaches allow the observation of individual DsRed mitochondria; however, the complexity of their 3D arrangement, combined with the limited space between mitochondria (as seen in Author response image 6), precludes a reliable estimation of mitochondrial organization within the midpiece. To overcome these challenges, we decided to study the midpiece architecture via SEM experiments on non-capacitated versus capacitated sperm stimulated with ionomycin to undergo the AE.

**Author response image 7. sa3fig7:** Organization of mitochondria observed via FF-SRM and SIM. Scale bar 2 µm. F.N:Fluorescence normalized. F:Frequency

Author response image 8 presents a single-cell comparison of the midpiece architecture in noncapacitated (NC) and acrosome-intact (AI) versus acrosome-reacted (AR) sperm, along with measurements of the midpiece diameter throughout its length. Notably, the diameter of the midpiece increases from the base of the head to more distal regions, ranging from 0.45 nm to 1.10 µm (as shown in Author response images 7 and 8). A significant correlation between the diameter of the flagellum and its curvature was observed (Author response image 9), suggesting a reorganization of the midpiece due to shearing forces. This is further exemplified in Author response images 8 and 9, which provide individual examples of this phenomenon.

**Author response image 8. sa3fig8:** Comparison of the midpiece architecture in acrosome-intact and acrosome-reacted sperm using scanning electron microscopy (SEM).

As expected, the overall diameter of the midpiece in AI sperm was larger than in AR sperm, with measurements of 0.731 ± 0.008 µm for AI and 0.694 ± 0.007 µm for AR (p = 0.013, Kruskal-Wallis test n > 100, N = 2), as shown in Author response image 10. Additionally, this Author response image 7 indicates that the reorganization of the midpiece architecture involves a change in the periodicity of the mitochondrial network, with frequencies shifting from fNC to fEA mitochondria per micron.

**Author response image 9. sa3fig9:** Comparison of the midpiece architecture in acrosome-intact (A) and acrosome-reacted (B) sperm using scanning electron microscopy (SEM).

Collectively, the structural results presented in Figure 5 and Author response images 6 to 10 demonstrate that the AE involves a comprehensive reorganization of the midpiece, affecting its diameter, pitch, and the organization of both the actin and mitochondrial networks. All this information is now incorporated in the new version of the paper (Figure. 2F)

**Author response image 10. sa3fig10:** Quantification of the midpiece diameter of the sperm flagellum in acrosome-intact and acrosome-reacted sperm analyzed by scanning electron microscopy (SEM). Data is presented as mean ± SEM. Kruskal-Wallis test was employed, p = 0.013 (AI n = 85 , AR n = 72).

(6) In the TG sperm used, the green fluorescence of the acrosome disappears when sperm die. Figure 1C should be analyzed only with live sperm by checking viability with propidium iodide or other means.

Response P2.6: We concur with Reviewer 2 that ideally, any experiment conducted for this study should include an intrinsic cell viability test. However, the current research employs a wide array of multidimensional imaging techniques that are not always compatible with, or might be suboptimal for, simultaneous viability assessments. In agreement with the reviewer's concerns, it is recognized that the data presented in Figure 1C may inherently be biased due to cell death. Nonetheless, Author response image 1 demonstrates that the relationship between AE and cell death is more complex than a straightforward all-or-nothing scenario. Specifically, Author response image 1C illustrates a case where the plasma membrane is compromised (Sytox Blue+) yet maintains acrosomal integrity (EGFP+). This observation contradicts Reviewer 1's assertion that "the green fluorescence of the acrosome disappears when sperm die," as discussed more comprehensively in response P2.3.

In light of these observations, we have meticulously revisited the entire manuscript to address and clarify potential biases in our results due to cell death. Consequently, Author response image 5 and its detailed description have been incorporated into the supplementary material of the manuscript to contribute to the transparency and reliability of our findings.

**Reviewer #3 (Public Review):**
(1) While progressive and also hyperactivated motility are required for sperm to reach the site of fertilization and to penetrate the oocyte's outer vestments, during fusion with the oocyte's plasma membrane it has been observed that sperm motility ceases. Identifying the underlying molecular mechanisms would provide novel insights into a crucial but mostly overlooked physiological change during the sperm's life cycle. In this publication, the authors aim to provide evidence that the helical actin structure surrounding the sperm mitochondria in the midpiece plays a role in regulating sperm motility, specifically the motility arrest during sperm fusion but also during earlier cessation of motility in a subpopulation of sperm post acrosomal exocytosis. The main observation the authors make is that in a subpopulation of sperm undergoing acrosomal exocytosis and sperm that fuse with the plasma membrane of the oocyte display a decrease in midpiece parameter due to a 200 nm shift of the plasma membrane towards the actin helix. The authors show the decrease in midpiece diameter via various microscopy techniques all based on membrane dyes, bright-field images and other orthogonal approaches like electron microscopy would confirm those observations if true but are missing. The lack of additional experimental evidence and the fact that the authors simultaneously observe an increase in membrane dye fluorescence suggests that the membrane dyes instead might be internalized and are now staining intracellular membranes, creating a false-positive result. The authors also propose that the midpiece diameter decrease is driven by changes in sperm intracellular Ca2+ and structural changes of the actin helix network. Important controls and additional experiments are needed to prove that the events observed by the authors are causally dependent and not simply a result of sperm cells dying.

Response P3.1: We appreciate the reviewer's observations and critiques. In response, we have expanded our experimental approach to include alternative methodologies such as mathematical modeling and electron microscopy, alongside further fluorescence microscopy studies. This diversified approach aims to mitigate potential interpretation artifacts and substantiate the validity of our observations regarding the contraction of the sperm midpiece. Additionally, we have implemented further control experiments to fortify the credibility and robustness of our findings, ensuring a more comprehensive and reliable set of results.

First, we acknowledge the concerns raised by Reviewer 2 regarding the interpretation of the magnitude of the observed contraction of the sperm flagellum's midpiece (see response P2.5). Specifically, we believe that the assertion that "... there is a decrease in midpiece parameter due to a 200 nm shift of the plasma membrane towards the actin helix" stated by reviewer 3 needs careful examination. We recognize that the fluorescence microscopy data provided might not conclusively support such a substantial shift. Our live cell imaging and superresolution microscopy experiments indicate that there is a significant decrease in the diameter of the sperm flagellum associated with AE. This is supported by colocalization experiments where FM4-64-stained structures (fluorescing upon binding to membranes) are observed moving closer to Sir-Actinlabeled structures (binding to F-actin). Quantitatively, Figure S5 describes the spatial shift between FM4-64 and Sir-Actin signals, narrowing from a range of 140-210 nm to 50-110 nm (considering the 2nd and 3rd quartiles of the distributions). The mean separation distance between both signals changes from 180 nm in AI cells to 70 nm in AR cells, a net shift of 110 nm. This observation suggests caution regarding the claim of a "200 nm shift of the plasma membrane towards the actin cortex."

Moreover, the concerns raised by Reviewer #3 about the potential internalization of membrane dyes, which might create a false-positive result by staining intracellular membranes, offer an alternative mechanism to explain a shift of up to 100 nm. This perspective is also supported by the critique from Reviewer #2 regarding the substantial distance (about 100 nm) between the plasma membrane and mitochondria post-acrosome reaction: “The authors state that the distance between the plasma membrane and mitochondria approaches about 100 nm after the acrosome reaction (…), but this is a very long distance and large structural changes may occur in the midpiece”. These insights have prompted us to refine our methodology and interpretation of the data to ensure a more accurate representation of the underlying biological processes.

Author response image 11 shows a first principles approach in two spatial dimensions to explore three scenarios where a membrane dye, such as FM4-64, stains structures at and within the midpiece of a sperm flagellum, but yet does not result in a net change of diameter. Author response image 11A-C illustrates three theoretical arrangements of fluorescent dyes: Model 1 features two rigid, parallel structures that mimic the plasma membrane surrounding the midpiece of the flagellum. Model 2 builds on Model 1 by incorporating the possibility of dye internalization into structures located near the membrane, suggesting a slightly more complex interaction with nearby membranous intracellular structures. Model 3 represents an extreme scenario where the fluorescent dyes stain both the plasma membrane and internal structures, such as mitochondrial membranes, indicating extensive dye penetration and binding. Author response image 11D-F displays the convolution of the theoretical fluorescent signals from Models 1 to 3 with the theoretical point spread function (PSF) of a fluorescent microscope, represented by a Gaussian-like PSF with a sigma of 19 pixels (approximately 300 nm). This process simulates how each model's fluorescence would manifest under microscopic observation, showing subtle differences in the spatial distribution of fluorescence among the models. Author response image 11G-I reveals the superresolution images obtained through Mean Shift Super Resolution (MSSR) processing of the models depicted in Author response image 11D-F.

By analyzing the three scenarios, it becomes clear that the signals from Models 2 and 3 shift towards the center compared to Model 1, as depicted in Author response image 11J. This shift in fluorescence suggests that the internalization of the dye and its interaction with internal structures might significantly influence the perceived spatial distribution and intensity of fluorescence, thereby impacting the interpretation of structural changes within the midpiece. Consequently, the experimentally observed contraction of up to 100 nm in could represent an actual contraction of the sperm flagellum's midpiece, a relocalization of the FM4-64 membrane dyes to internal structures, or a combination of both scenarios.

To discern between these possibilities, we implemented a scanning electron microscopy (SEM) approach. The findings presented in Figure 5 and Author response images 7 to 9 conclusively demonstrate that the AE involves a comprehensive reorganization of the midpiece. This reorganization affects its diameter, which changes by approximately 50 nm, as well as the pitch and the organization of both the actin and mitochondrial networks. This data corroborates the structural alterations observed and supports the validity of our interpretations regarding midpiece dynamics during the AE.

**Author response image 11. sa3fig11:** Modeling three scenarios of midpiece staining with membrane fluorescent dyes.

Secondly, we wish to clarify that in some of our experiments, we have utilized changes in the intensity of FM4-64 fluorescence as an indirect measure of midpiece contraction. This approach is supported by a linear inverse correlation between these variables, as illustrated in Figure S2D. It is important to note that this observation is correlative and indirect; therefore, our data does not directly substantiate the claim that "in a subpopulation of sperm undergoing AE and sperm that fuse with the plasma membrane of the oocyte, there is a decrease in midpiece parameter due to a 200 nm shift of the plasma membrane towards the actin helix". Specifically, we have not directly measured the distance between the plasma membrane and actin cortex in experiments involving gamete fusion.

All the concerns highlighted in this Response P1.1 have been addressed and incorporated into the manuscript. This addition aims to provide comprehensive insight into the experimental observations and methodologies used, ensuring that the data is transparent and accessible for thorough review and replication.

**Editor Comment:**
As the authors can see from the reviews, the reviewers had quite different degrees of enthusiasm, thus discussed extensively. The major points in consensus are summarized below and it is highly recommended that the authors consider their revisions.(1) Causality of midpiece contraction with motility arrest is not conclusively supported by the current evidence. Time-resolved imaging of FM4-64 and motility is needed and the working model needs to be revised with two scenarios - whether the sperm contracting indicates a fertilizing sperm or sperm to be degenerated.(2) The rationale for using FM4-64 as a plasma membrane marker is not clear as it is typically used as an endo-membrane marker, which is also related to the discrepancy of Fluo-4 signal diameter vs. FM4-64 (Figure 4E). The viability of sperm with increased FM4-64 needs to be demonstrated.(3) The mechanism of midpiece contraction in motility cessation along the whole flagellum is not discussed.(4) The use of an independent method to support the changes in midpiece diameter/structural changes such as DsRed (transgenic) or TEM.(5) The claim of Ca2+ change needs to be toned down.

Response Editor: We thank the editor and the reviewers for their thorough and positive assessment of our work and the constructive feedback to further improve our manuscript. Please find below our responses to the reviewers’ comments. We have addressed all these points in the current version. Briefly,

(1) Time resolved images to show the correlation between FM4-64 fluorescence increase and the motility was incorporated

(2) The rationale for using FM4-64 was added.

(3) The mechanism of midpiece contraction was discussed in the paper

(4) An independent method was included to support our conclusions (SEM and other markers not based on membrane dyes)

(5) The results related to the calcium increase were toned down.

**Recommendations for the authors:**

**Reviewer #1 (Recommendations For The Authors):**
(1) To claim midpiece actin polymerization/re-organization is required for AE, demonstrating that AE does not occur in the presence of actin depolymerizing drugs (e.g., Latrunculin A, Cytochalasin D) would be necessary since the current data only shows the association/correlation. Was the block of AE by actin depolymerization observed?

Response R1.1: We agree with the reviewer but unfortunately, since actin polymerization and or depolymerization in the head are important for exocytosis, we cannot use this experimental approach to dissect both events. Addition of these inhibitors block the occurrence of AE (PMID: 12604633).

(2) Please provide the rationale for using FM4-64 to visualize the plasma membrane since it has been reported to selectively stain membranes of vacuolar organelles. What is the principle of increase of FM4-64 dye intensity, other than the correlation with midpiece contraction? For example, in lines 400-402: the authors mentioned that 'some acrosomereacted moving sperm within the perivitelline space had low FM4-64 fluorescence in the midpiece (Figure 6C). After 20 minutes, these sperm stopped moving and exhibited increased FM4-64 fluorescence, indicating midpiece contraction (Figure 6D).' While recognizing the increase of FM4-64 dye intensity can be an indicator of midpiece contraction, without knowing how and when the intensity of FM4-64 dye changes, it is hard to understand this observation. Please discuss.

Response R1.2: FM4-64 is an amphiphilic styryl fluorescent dye that preferentially binds to the phospholipid components of cell membranes, embedding itself in the lipid bilayer where it interacts with phospholipid head groups. Due to its amphiphilic nature, FM dyes primarily anchor to the outer leaflet of the bilayer, which restricts their internalization. It has been demonstrated that FM4-64 enters cells through endocytic pathways, making these dyes valuable tools for studying endocytosis.

Upon binding, FM4-64's fluorescence intensifies in a more hydrophobic environment that restricts molecular rotation, thus reducing non-radiative energy loss and enhancing fluorescence. These photophysical properties render FM dyes useful for observing membrane fusion events. When present in the extracellular medium, FM dyes rapidly reach a chemical equilibrium and label the plasma membrane in proportion to the availability of binding sites.

In wound healing studies, for instance, the fluorescence of FM4-64 is known to increase at the wound site. This increase is attributed to the repair mechanisms that promote the fusion of intracellular membranes at the site of the wound, leading to a rise in FM4-64 fluorescence. Similarly, an increase in FM4-64 fluorescence has been reported in the heads of both human and mouse sperm, coinciding with AE. In this scenario, the fusion between the plasma membrane and the acrosomal vesicle provides additional binding sites for FM4-64, thus increasing the total fluorescence observed in the head. This dynamic response of FM4-64 makes it an excellent marker for studying these cellular processes in real-time.

This study is the first to report an increase in FM4-64 fluorescence in the midpiece of the sperm flagellum. Figures 5 and Author response images 6 to 9 demonstrate that during the contraction of the sperm flagellum, structural rearrangements occur, including the compaction of the mitochondrial sheath and other membranous structures. Such contraction likely increases the local density of membrane lipids, thereby elevating the local concentration of FM4-64 and enhancing the probability of fluorescence emission. Additionally, changes in the microenvironment such as pH or ionic strength during contraction might further influence FM4-64’s fluorescence properties, as detailed by Smith et al. in the Journal of Membrane Biology (2010). The photophysical behavior of FM4-64, including changes in quantum yield due to tighter membrane packing or alterations in curvature or tension, may also contribute to the increased fluorescence observed. Notably, Figure S2 indicates that other fluorescent dyes like Memglow 700, Bodipy-GM, and FM1-43 also show a dramatic increase in their fluorescence during the midpiece contraction. Investigating whether the compaction of the plasma membrane or other mesoscale processes occur in the midpiece of the sperm flagellum could be a valuable area for future research. The use of fluorescent dyes such as LAURDAN or Nile Red might provide further insights into these membrane dynamics, offering a more comprehensive understanding of the biochemical and structural changes during sperm motility and gamete fusion events.

(3) As the volume of the whole midpiece stays the same while the diameter decreases along the whole midpiece (midpiece contraction), the authors need to describe what changes in the midpiece length they observe during the contraction. Was the length of the midpiece during the contraction measured and compared before and after contraction?

Response R1.3: As requested, we have measured the length of the midpiece in AI and AR sperm. As shown in Author response image 12 (For review purposes only), no statistically significant differences were observed.

**Author response image 12. sa3fig12:** Midpiece length measured by the length of mitochondrial DsRed2 fluorescence in EGFP-DsRed2 sperm. Measurements were done before (acrosome-intact) and after (acrosome-reacted) acrosome exocytosis and midpiece contraction. Data is presented as the mean ± sem of 14 cells induced by 10 µM ionomycin. Paired t-test was performed, resulting in no statistical significance.

(4) Most of all, it is not clear what the midpiece, thus mitochondria, contraction means in terms of sperm bioenergetics and motility cessation. Would the contraction induce mitochondrial depolarization or hyperpolarization, increase or decrease of ATP production/consumption? It will be great if this point is discussed. For example, an increase in mitochondrial Ca2+ is a good indicator of mitochondrial activity (ATP production).

Response R1.4: That is an excellent point. We have discussed this idea in the discussion (line 620-624). We are currently exploring this idea using different approaches because we also think that these changes in the midpiece may have an impact in the function of the mitochondria and perhaps, in their fate once they are incorporated in the egg after fertilization.

(5) The authors claimed that Ca2+ signal propagates from head to tail, which is the opposite of the previous study (PMID: 17554080). Please clarify if it is a speculation. Otherwise, please support this claim with direct experimental evidence (e.g., high-speed calcium imaging of single cells).

Response R1.5: In that study, it was claimed that a [Ca2+]i increase that propagates from the tail to the head occurs when CatSper is stimulated. They did not evaluate the occurrence of AE when monitoring calcium.

Our data is in agreement with our previous results (PMID: 26819478) that consistently indicated that only the[Ca2+]i rise originating in the sperm head is able to promote AE.

(6) Figure 4E: Please explain how come Fluo4 signal diameter can be smaller than FM4-64 dye if it stains plasma membrane (at 4' and 7').

Response R1.6: When colocalizing a diffraction-limited image (Fluo4) with a super-resolution image (FM4-64), discrepancies in signal sizes and locations can become apparent due to differences in resolution. The Fluo4 signal, being diffraction-limited, adheres to a resolution limit of approximately 200-300 nanometers under conventional light microscopy. This limitation causes the fluorescence signal to appear broader and less defined. Conversely, super-resolution microscopy techniques, such as SRRF (Super-Resolution Radial Fluctuations), achieve resolutions down to tens of nanometers, allowing FM4-64 to reveal finer details at the plasma membrane and display potentially smaller apparent sizes of stained structures. Although both dyes might localize to the same cellular regions, the higher resolution of the FM4-64 image allows it to show a more precise and smaller diameter of the midpiece of the flagellum compared to the broader, less defined signal of Fluo4. To address this, the legend of Figure 4E has been slightly modified to clarify that the FM4-64 image possesses greater resolution.

(7) Figure 5D-G: the midpiece diameter of AR intact cells was shown ~ 0.8 um or more in Figure 2, while now the radius in Figure 5 is only 300 nm. Since the diameter of the whole midpiece is nearly uniform when the acrosome is intact, clarify how and what brings this difference and where the diameter/radius measurement is done in each figure.

Response R1.7: The difference resides in what is being measured. In Figure 2, the total diameter of the cell is measured, through the maximum peaks of FM4-64 fluorescence which is a probe against plasma membrane. As for Figure 5, the radius shown makes reference to the radius of the actin double helix within the midpiece. To that end, cells were fixed and stained with phalloidin, a F-actin probe.

Minor points(8) Figure S1 title needs to be changed. The "Midpiece contraction" concept is not introduced when Figure S1 is referred to.

Response R1.8: This was corrected in the new version.

(9) Reference #19: the authors are duplicated.

Response R1.9: This was corrected in the new version.

(10) Line 315-318: sperm undergoing contraction -> sperm undergoing AR/AE?

Response R1.10: This was corrected in the new version.

(11) Line 3632 -> punctuation missing.

Response R1.11: Modified as requested.

(12) Movie S7: please add an arrow to indicate the spermatozoon of interest.

Response R1.12: The arrow was added as suggested.

(13) Line 515: One result of this study was that the sperm flagellum folds back during fusion coincident with the decrease in the midpiece diameter. The authors did not provide an explanation for this observation. Please speculate the function of this folding for the fertilization process.

Response R1.13: As requested, this is now incorporated in the discussion. We speculate that the folding of the flagellum during fusion further facilitates sperm immobilization because it makes it more difficult for the flagellum to beat. Such processes can enhance stability and increase the probability of fusion success. Mechanistically, the folding may occur as a consequence of the deformation-induced stress that develops during the decrease of midpiece diameter.

**Reviewer #2 (Recommendations For The Authors):**
(1) Figure 2C, D, E. Does "-1" on the X-axis mean one minute before induction? If so, the diameter is already smaller and FM4-64 fluorescence intensity is higher before the induction in the spontaneous group. Does the acrosome reaction already occur at "-1" in this group?

Response R2.1: Yes, “-1” means that the measurements of the diameter/FM4-64 fluorescence was done one minute before the induction. And it is correct that the diameter is smaller and FM464 fluorescence higher in the spontaneous group because these sperm underwent acrosome exocytosis before the induction, that is, spontaneously.

(2) Figure 3D. Purple dots are not shown in the graph on the right side.

Response R2.2: Modified as requested.

(3) Lines 404-406. "These results suggest that midpiece contraction and motility cessation occur only after acrosome-reacted sperm penetrate the zona pellucida". Since midpiece contraction and motility cessation also occur before the passage through the zona pellucida (Figure 9B), "only" should be deleted.

Response R2.3: Modified as requested.

**Reviewer #3 (Recommendations For The Authors):**
(1) Do the authors have a hypothesis as to why the observed decrease in midpiece parameter results in cessation of sperm motility? It would be beneficial for the manuscript to include a paragraph about potential mechanisms in the discussion.

Response R3.1: As requested, a potential mechanism has been proposed in the discussion section (line 644-656).

(2) Since the authors propose in Gervasi et al. 2018 that the actin helix might be responsible for the integrity of the mitochondrial sheath and the localization of the mitochondria, is it possible that the proposed change in plasma membrane diameter and actin helix remodeling for example alters the localization of the mitochondria? TEM should be able to reveal any associated structural changes. In its current state, the manuscript lacks experimental evidence supporting the author's claim that the "helical actin structure plays a role in the final stages of motility regulation". The authors should either include additional evidence supporting their hypothesis or tone down their conclusions in the introduction and discussion.

Response R3.3: We agree with the reviewer. This is an excellent point. As suggested by this reviewer as well as the other reviewers, we have performed SEM to observe the changes in the midpiece observed after its contraction for two main reasons. First, to confirm this observation using a different approach that does not involve the use of membrane dyes. As shown in Author response image 6-10, we have observed that in addition to the midpiece diameter, there is a reorganization of the mitochondria sheet that is also suggested by the SIM experiments. These observations will be explored with more experiments to confirm the structural and functional changes that mitochondria undergo during the contraction. We are currently investigating this phenomenon, These results are now included in the new Figure 2F.

(3) In line 134: The authors write: 'Some of the acrosome reacted sperm moved normally, whereas the majority remained immotile". Do the authors mean that a proportion of the sperm was motile prior to acrosomal exocytosis and became immotile after, or were the sperm immotile to begin with? Please clarify.

Response R3.4: This statement is based on the quantification of the motile sperm after induction of AE within the AR population (Fig. 1C).

(4) The authors do not provide any experimental evidence supporting the first scenario. In video 1 a lot of sperm do not seem to be moving to begin with, only a few sperm show clear beating in and out of the focal plane. The highlighted sperm that acrosome-reacted upon exposure to progesterone don't seem to be moving prior to the addition of progesterone. In contrast, the sperm that spontaneously acrosome react move the whole time. In video 1 this reviewer was not able to identify one sperm that stopped moving upon acrosomal exocytosis. Similarly in video 3, although the resolution of the video makes it difficult to distinguish motile from non-motile sperm. In video 2 the authors only show sperm that are already acrosome reacted. Please explain and provide additional evidence and statistical analysis supporting that sperm stop moving upon acrosomal exocytosis.

Response R3.5: In videos 1 and 3, the cells are attached to the glass with concanavalin-A, this lectin makes sperm immotile (if well attached) because both the head and tail stick to the glass. The observed motility of sperm in these videos is likely due to them not being properly attached to the glass, which is completely normal. On the contrary, in videos 2 and 4, sperm are attached to the glass with laminin. This is a glycoprotein that only binds the sperm to the glass through its head, that is why they move freely.

(5) Could the authors provide additional information about the FM4-64 fluorescent dye?What is the mechanism, and how does it visualize structural changes at the flagellum? Since the whole head lights up, does that mean that the dye is internalized and now stains additional membranes, similar to during wound healing assays (PMID 20442251, 33667528). Or is that an imaging artifact? How do the authors explain the correlation between FM4-64 fluorescence increase in the midpiece and the observed change in diameter? Does FM4-64 have solvatochromatic properties?

Response R3.6: We appreciate the insightful queries posed by Reviewer 3, which echo the concerns initially brought forward by Reviewer 1. For a detailed explanation of the mechanism of FM4-64 dye, how we interpret it, visualizes structural changes in the flagellum, and its behavior during cellular processes, please refer to our detailed response in Response R1.2. In brief, FM464 is a lipophilic styryl dye that preferentially binds to the outer leaflets of cellular membranes due to its amphiphilic nature. Upon binding, the dye becomes fluorescent, allowing for the visualization of membrane dynamics. The increase in fluorescence in the sperm head or midpiece likely results from the dye’s accumulation in areas where membrane restructuring occurs, such as during AE or in response to changes in the flagellum structure.

Regarding the specific questions about internalization and whether FM4-64 stains additional membranes similarly to what is observed in wound healing assays, it's important to note that FM4-64 can indeed be internalized through endocytosis and subsequently label internal vesicular structures. Additionally, FM4-64 may experience changes in its fluorescence as a result of fusion events that increase the lipid content of the plasma membrane, as observed in studies cited (PMID 20442251, 33667528). This characteristic makes FM4-64 valuable not only for outlining cell membranes but also for tracking the dynamics of both internal and external membrane systems, particularly during cellular events that involve significant membrane remodeling, such as wound healing or AE.

Concerning whether the increased fluorescence and observed changes in diameter are artifacts or reflect real biological processes, the correlation observed likely indicates actual changes in the midpiece architecture through molecular mechanisms that remain to be further elucidated. The data presented in Figures 5 and Author response images 6-10 support that this increase in fluorescence is not merely an artifact but a feature of how FM4-64 interacts with its environment.

Finally, regarding the solvatochromatic properties of FM4-64, while the dye does show changes in its fluorescence intensity in different environments, its solvatochromatic properties are generally less pronounced than those of dyes specifically designed to be solvatochromatic. FM464's fluorescence changes are more a result of membrane interaction dynamics and dye concentration than of solvatochromatic shifts.

(6) For the experiment summarized in Figure S1, did the authors detect sperm that acrosome-reacted upon exposure to progesterone and kept moving? This reviewer is wondering how the authors reliably measure FM4-64 fluorescence if the flagellum moves in and out of the focal plane. If the authors observe sperm that keep moving, what was the percentage within a sperm population and how did FM4-64 fluorescence change?

Response R3.6: We did identify sperm that underwent acrosome reaction upon exposure to progesterone and continued to exhibit movement. However, due to the issue raised by the reviewer regarding the flagellum going out of focus, we opted to quantify the percentage of sperm that were adhered to the slide (using laminin). This approach allows for the observation of flagellar position over time, facilitating an easy assessment of fluorescence changes. The percentage of sperm that maintained movement after AE is depicted in Figure 1C.

(7) In Figure S1B it doesn't look like the same sperm is shown in all channels or time points, the hook shown in the EGFP channel is not always pointing in the same direction. If FM4-64 is staining the plasma membrane, how do the authors explain that the flagellum seems to be more narrow in the FM4-64 channel than in the brightfield and DsRed2 channel?

Response 3.7: It is the same sperm, but due to technical limitations images were sequentially acquired. For example, for time 5 minutes after progesterone, all images in DIC were taken, then all images in the EGFP channel, then DsRed2* and finally FM4-64. The reason for this was to acquire images as fast as possible, particularly in DIC images which were then processed to get the beat frequency.

Regarding the flagellum that seems to be more narrow in the FM4-64 channel compared to the BF or DsRed2 channel, the explanation is related to the fact that intensity of the DsRed2 signal is stronger than the other two. This higher signal may have increased the amount of photons captured by the detector.

(8) Overall, it would be beneficial to include statistics on how many sperm within a population did change FM4-64 fluorescence during AE and how many did not, in addition to information about motility changes and viability. Did the authors exclude that the addition of FM4-64 causes cell death which could result in immotile sperm or that only dying sperm show an increase in FM4-64 fluorescence?

Response 3.8: The relationship between cell death and the increase in FM4-64 fluorescence is widely discussed in Response P2.3. In our experiments, we always considered sperm that were motile to hypothesize about the relevance of this observation. We have two types of experiments:

(1) Sperm-egg Fusion: In experiments where sperm and eggs were imaged to observe their fusion, sperm were initially moving and after fusion, the midpiece contraction (increase in FM4-64 fluorescence was observed) indicating that the change in the midpiece (that was observed consistently in all fusing cells analyzed), is part of the process.

(2) Sperm that underwent AE: we have observed two behaviours as shown in Figure 1:

a) Sperm that underwent AE and they remain motile without midpiece contraction (they are alive for sure);

b) Sperm that underwent AE and stopped moving with an increase in FM464 fluorescence. We propose that this contraction during AE is not desired because it will impede sperm from moving forward to the fertilization site when they are in the female reproductive tract. In this case, we acknowledge that the cessation of sperm motility may be attributed to cellular death, potentially correlating with the increased FM4-64 signal observed in the midpiece of immotile sperm that have undergone AE. To address this hypothesis, we conducted image-based flow cytometry experiments, which are well-suited for assessing cellular heterogeneity within large populations.

Regarding the relationship between the increase in FM4-64 and AE, we have always observed that AE is followed by an increase in FM4-64 in the head in mice (PMID: 26819478) as well as in human (PMID: 25100708) sperm. This was originally corroborated with the EGFP sperm. However, not all the cells that undergo AE increase the FM4-64 fluorescence in the midpiece.

(9) The authors report that a fraction of sperm undergoes AE without a change in FM4-64 fluorescence (Figure 1F). How does the [Ca2+]i change in those cells? Again statistics on the distribution of a certain pattern within a population in addition to showing individual examples would be very helpful.

Response 3.9: A recent work shows that an initial increase in [Ca2+]i is required to induce changes in flagellar beating necessary for hyperactivation (Sánchez-Cárdenas et al., 2018). However, when [Ca2+]i increases beyond a certain threshold, flagellar motility ceases. These conclusions are based on single-cell experiments in murine sperm with different concentrations of the Ca2+ ionophore, A23187. The authors reported that complete loss of motility was observed when using ionophore concentrations higher than 1 μM. In contrast, spermatozoa incubated with 0.5 μM A23187 remained motile throughout the experiment. Once the Ca2+ ionophore is removed, the sperm would reduce the concentration of this ion to levels compatible with motility and hyperactivation (Navarrete et al., 2016). However, some of the washed cells did not recover mobility in the recorded time window (Sánchez-Cárdenas et al., 2018). These results would indicate that due to the increase in [Ca2+]i induced by the ionophore, irreversible changes occurred in the sperm flagellum that prevented recovery of mobility, even when the ionophore was not present in the recording medium.

Taking into account our results, one possible scenario to explain this irreversible change would be the contraction of the midpiece. Our results demonstrate that the increase in [Ca2+]i observed in the midpiece (whether by induction with progesterone, ionomycin or occurring spontaneously) causes the contraction of this section of the flagellum and its subsequent immobilization.

(10) While the authors results show that changes in [Ca2+]i correlate with the observed reduction of the midpiece diameter, they do not provide evidence that the structural changes are triggered by Ca2+i influx. It could just be a coincidence that both events spatially overlap and that they temporarily follow each other. The authors should either provide additional evidence or tone down their conclusion.

Response 3.10: We agree with the reviewer. As suggested, we have toned down our conclusion.

(11) Are the authors able to detect the changes in the midpiece diameter independent from FM4-64 or other plasma membrane dyes? An alternative explanation could be that the dyes are internalized due to cell death and instead of staining the plasma membrane they are now staining intracellular membranes, resulting in increased fluorescence and giving the illusion that the midpiece diameter decreased. How do the authors explain that the Bodipy-GM1 Signal directly overlaps with DsRed2 and SIR-actin, shouldn't there be some gap? Since the rest of the manuscript is based on that proposed decrease in midpiece diameter the authors should perform orthogonal experiments to confirm their observation.

Response 3.11: As requested by the reviewer, we have not used new methods to visualize the change in sperm diameter in the midpiece. In neither of them, a membrane dye was used. First, we have performed immunofluorescence to detect a membrane protein (GLUT3). Second, we have used scanning electron microscopy. The results are now incorporated in the new Figure 2FG. In both experiments, a change in the midpiece diameter was observed. Please, also visit responses P2.5 and Author response images 8 to 10.

Regarding the overlap between the signal of Bodipy GM1 (membrane) and the fluorescence of DsRed2 (mitochondria) and Sir-Actin (F-actin), it is only observed in acrosomereacted sperm, not in acrosome-intact sperm (Figure S4). In our view, these structures become closed after midpiece contraction, and the resolution of the images is insufficient to distinguish them clearly. This issue is also evident in Figure 5B. Therefore, we conducted additional experiments using more powerful super-resolution techniques such as STORM (Figures 5D-F).

(12) The proposed gap of 200 nM between the actin helix and the plasma membrane, has been observed by TEM? Considering that the diameter of the mouse sperm midpiece is about 1 um, that is a lot of empty space which leaves only about 600 nm for the rest of the flagellum. The axoneme is 300 nm and there needs to be room for the ODFs and the mitochondria. Please explain.

Response 3.12: Unfortunately, the filament of polymerized actin cannot be observed by TEM. Furthermore, we were discouraged from trying other approaches, such as utilizing phalloidin gold, because for some reason, it does not work properly.

In our view, the 200 nm gap between the actin cytoskeleton and the plasma membrane is occupied by the mitochondria (that is the size that it is frequently reported based on TEM; see https://doi.org/10.1172/jci.insight.166869).

(13) The results provided by the authors do not convince this reviewer that the actin helix moves, either closer to the plasma membrane or toward the mitochondria, the observed differences are minor and not confirmed by statistical analysis.

Response 3.13: As requested, the title of that section was changed. Moreover, our conclusion is exactly as the reviewer is suggesting: “Since the results of the analysis of SiR-actin slopes were not conclusive, we studied the actin cytoskeleton structure in more detail”. This conclusion is based on the statistical analysis shown in Figure S5D-E.

(14) The fluorescence intensity of all plasma membrane dyes increases in all cells chosen by the authors for further analysis. Could the increase in SiR-Actin fluorescence be explained by a microscopy artifact instead of actin helix remodeling? Alternatively, can the authors exclude that the observed increase in SIR-Actin might be an artifact caused by the increase in FM4-64 fluorescence? Since the brightness in the head similarly increases to the fluorescence in the flagellum the staining pattern looks suspiciously similar. Did the authors perform single-stain controls?

Response 3.14: We had similar concerns when we were doing the experiments using SiR-actin. Although we have performed single stain controls to make sure that the actin helix remodelling occurs during the midpiece contraction, we have performed experiments using higher resolution techniques such as STORM using a different probe to stain actin (Phalloidin).

(15) Should actin cytoskeleton remodeling indeed result in a decrease of actin helix diameter, what do the authors propose is the underlying mechanism? Shouldn't that result in changes in mitochondrial structure or location and be visible by TEM? This reviewer is also wondering why the authors focus so much on the actin helix, while the plasma membrane based on the author's results is moving way more dramatically.

Response 3.15: This raises an intriguing point. Currently, we lack an understanding of the underlying mechanism driving actin remodeling, and we are eager to conduct further experiments to explore this aspect. For instance, we are investigating the potential role of Cofilin in remodeling the F-actin network. Initial experiments utilizing STORM imaging have revealed the localization of Cofilin in the midpiece region, where the actin helix is situated.

Regarding mitochondria, thus far, we have not uncovered any evidence suggesting that acrosome reaction or fusion with the egg induces a rearrangement of these organelles within the structure. The rationale for investigating polymerized actin in depth stems from the fact that, alongside the axoneme and other flagellar structures such as the outer dense fibers and fibrous sheet, these are the sole cytoskeletal components present in that particular tail region.

(14) The fact that the authors observe that most sperm passing through the zona pellucida, which requires motility, display high FM4-64 fluorescence, doesn't that contradict the authors' hypothesis that midpiece contraction and motility cessation are connected? Videos confirming sperm motility and information about pattern distribution within the observed sperm population in the perivitelline space should be provided.

Response 3.14: We believe it is a matter of time, as depicted in Figure 1D, our model shows that first the cells lose the acrosome, present motility and low FM4-64 fluorescence in the midpiece (pattern II) and after that, they lose motility and increase FM4-64 fluorescence in the midpiece (pattern III). That is why, we think that when sperm pass the zona pellucida they present pattern II and after some time they evolve into pattern III.

(15) In the experiments summarized in Figure 8, did all sperm stop moving? Considering that 74 % of the observed sperm did not display midpiece contraction upon fusion, again doesn't that contradict the authors' hypothesis that the two events are interdependent? Similarly, in earlier experiments, not all acrosome-reacted sperm display a decrease in midpiece diameter or stop moving, questioning the significance of the event. If some sperm display a decrease in midpiece diameter and some don't, or undergo that change earlier or later, what is the underlying mechanism of regulation? The observed events could similarly be explained by sperm death: Sperm are dying × plasma membrane integrity changes and plasma membrane dyes get internalized × [Ca2+]i simultaneously increases due to cell death × sperm stop moving.

Response 3.15: The percentage of sperm that did not exhibit midpiece contraction in Fig.8B is 26%, not 74%, indicating that it does not contradict our hypothesis. However, this still represents a significant portion of sperm that remain unchanged in the midpiece, leaving room for various explanations. For instance, it's possible that: (i) the change in fluorescence was not detected due to the event occurring after the recording concluded, or (ii) in some instances, this alteration simply does not occur. Nevertheless, we did not track subsequent events in the oocyte, such as egg activation, to definitively ascertain the success of fusion. Incorporation of the dye only manifests the initiation of the process.

(16) The authors propose changes in Ca2+ as one potential mechanism to regulate midpiece contraction, however, the Ca2+ measurements during fusion are flawed, as the authors write in the discussion, by potential Ca2+ fluorophore dilution. Considering that the authors observe high Ca2+ in all sperm prior to fusion, could that be a measuring artifact? Were acrosome-intact sperm imaged with the same settings to confirm that sperm with low and high Ca2+ can be distinguished? Should [Ca2+]i changes indeed be involved in the regulation of motility cessation during fusion, could the authors speculate on how [Ca2+]i changes can simultaneously be involved in the regulation of sperm hyperactivation?

Response 3.16: We agree with the reviewer that our experiments using calcium probes are not conclusive for many technical problems. We have toned down our conclusions in the new version of the manuscript.

(17) 74: AE takes place for most cells in the upper segment of the oviduct, not all of them.Please correct.

Response 3.17: Corrected in the new version.

(18) 88: Achieved through, or achieved by, please correct.

Response 3.18: Corrected in the new version.

(19) 243: Acrosomal exocytosis initiation by progesterone, please specify.

Response 3.19: Modified in the new version.

(20) 277: "The actin cytoskeleton approaches the plasma membrane during the contraction of the midpiece" is misleading. The author's results show the opposite.

Response 3.20: As suggested, this statement was modified.

(21) 298: Why do the authors find it surprising that the F-actin network was unchanged in acrosome-intact sperm that do not present a change in midpiece diameter?

Response 3.21: The reviewer is right. The sentence was modified.

(22) Figures 5D,F: The provided images do not support a shift in the actin helix diameter.

Response 3.22: The shift in the actin helix diameter is provided in Figure 5E and 5G.

(23) Figure S5C: The authors should show representative histograms of spontaneously-, progesterone induced-, and ionomycin-induced AE. Based on the quantification the SiRactin peaks don't seem to move when the AR is induced by progesterone.

Response 3.23: As requested, an ionomycin induced sperm is incorporated.

(24) 392: Which experimental evidence supports that statement?

Response 3.24: A reference was incorporated.

Reference 13 is published, please update. Response 3.25: updated as requested.